# Wind turbine main-bearing lubrication - Part 1: An introductory review of elastohydrodynamic lubrication theory

Edward Hart[1], Elisha de Mello[2], and Rob Dwyer-Joyce[2]

[1]Wind Energy and Control Centre, Department of Electronic and Electrical Engineering, The University of Strathclyde, Glasgow, UK

[2]Leonardo Centre for Tribology, Department of Mechanical Engineering, The University of Sheffield, UK

**Correspondence:** Edward Hart (edward.hart@strath.ac.uk)

**Abstract.** This paper is the first in a two-part study on lubrication in wind turbine main-bearings. Elastohydrodynamic lubrication is a complex field, the formulas and results from which should not be applied blindly, but with proper awareness and consideration of their context, validity and limitations in any given case. The current paper, "Part 1", therefore presents an introductory review of elastohydrodynamic lubrication theory in order to provide this necessary background and context in an accessible form, promoting cross-disciplinary understanding. Fundamental concepts, derivations and formulas are presented, followed by the more advanced topics of: starvation, non-steady effects, surface roughness interactions and grease lubrication. "Part 2" applies the presented material in order to analyse wind turbine main-bearing lubrication in the context of available film thickness formulas and related results from lubrication theory. Aside from the main-bearing, the material presented here is also applicable to other lubricated non-conformal contacts in wind turbines, including pitch and yaw bearings and gear-teeth.

## 1 Introduction

Wind turbine main-bearings have come under increased research scrutiny of late, due to higher than expected failures rates and failure mechanisms which are yet to be fully understood (Hart et al., 2019, 2020; Hart, 2020; Guo et al., 2021; Nejad et al., 2022). Integral to main-bearing function and performance is the fact that it is a rolling bearing, tasked with allowing low-friction, free rotation of the shaft while also supporting the turbine rotor. Lubrication of the main-bearing is therefore necessary to prevent rapid wear and damage propagation from taking place. As such, the lubricant and lubrication mechanisms acting within this component must be accounted for as part of any attempt to fully characterize and understand main-bearing internal operational conditions, failure mechanisms and reliability. This two-part study seeks to begin this process.

Lubrication, and elastohydrodynamic lubrication (EHL) in particular, is a complex, nuanced and rapidly evolving field. While simplified film thickness formulas have been developed, their application should be accompanied by careful consideration of their validity and possible limitations in any given case. Furthermore, additional effects may be present where operational conditions vary rapidly or where grease lubrication is used. For the benefit of non-EHL-specialists, it is therefore argued that there is considerable value in an introductory review of this field which presents the reader with a comprehensive overview of EHL theory, including: fundamental equations and the problem formulation, the approximations being applied, numerical solution methods, general characteristics of EHL contacts, simplified film thickness equations and their validity, and,

an overview of additional effects caused by time-varying conditions, starvation, surface roughness and grease behaviour. While a number of excellent review papers are available in this field (Spikes, 2006; Lugt and Morales-Espejel, 2011; Greenwood, 2020), some are older now and for others a prior familiarity with topic fundamentals is ideally required. In the case of review papers which cover specific topics within EHL (Morales-Espejel and Wemekamp, 2008; Lugt, 2009, 2016; Morales-Espejel, 2014; Poll et al., 2019; Marian et al., 2020; Meng et al., 2020), a much higher level of familiarity is necessary. This paper, "Part 1" of the main-bearing lubrication study, therefore presents an introductory[1] review of EHL for non-conformal (roller bearings, gear-teeth *etc.*) contacts which covers both topic fundamentals and recent results, while remaining accessible to a more general engineering audience. While main-bearings form the focus of the overall study, the material presented in Part 1 applies equally to other lubricated non-conformal contacts in wind turbines, including pitch and yaw bearings and gear-teeth. To aid the reader, a table of symbols is provided in Appendix A.

## 2 Surface separation and lubrication regimes

Fluid film lubrication exists when two machine surfaces are completely separated by a layer of lubricant. In such circumstances, forces are carried via pressures generated within the lubricant and frictional/wear conditions are greatly improved. The presence of an adequate lubricant film is therefore critical to the reliability and longevity of machine components. In the context of wind energy, one normally encounters non-conformal contacts (roller bearings, gear-teeth *etc.*) operating in the elastohydrodynamic regime[2], in which significant elastic deformations of lubricated surfaces occur.

For completely smooth surfaces a lubricant film would always be present, even if vanishingly small. However, in reality material surfaces are not perfectly smooth and exhibit roughness, geometrical variations, on the order of 0.01-10 $\eta$m (Hamrock et al., 2004). It transpires that typical lubricating film thicknesses also sit somewhere within this range, meaning film thickness needs to be considered relative to surface roughness in order to determine whether a separation of surfaces has been achieved. The appropriate relative quantity is the film parameter,

$$\Lambda = \frac{h_m}{\sigma}, \tag{1}$$

which relates the minimum film thickness, $h_m$, to the combined (root-mean-square) roughness of contacting surfaces I and II, $\sigma = \sqrt{\sigma_{\mathrm{I}}^2 + \sigma_{\mathrm{II}}^2}$ (Hamrock et al., 2004). While delineations between lubrication regimes are difficult to make exactly, the following rough estimates indicate film parameter values associated with each (Hamrock et al., 2004):

- Hydrodynamic lubrication, $5 < \Lambda < 100$

- Elastohydrodynamic lubrication, $3 < \Lambda < 10$

- Mixed lubrication, $1 < \Lambda < 5$

- Boundary lubrication, $\Lambda < 1$.

---

[1]This being the case, the aim is to provide an accessible and representative overview of EHL theory, as opposed to an exhaustive review of the entire field.
[2]Although a novel (conformal) journal bearing design in this space is being developed (Rolink et al., 2020, 2021).

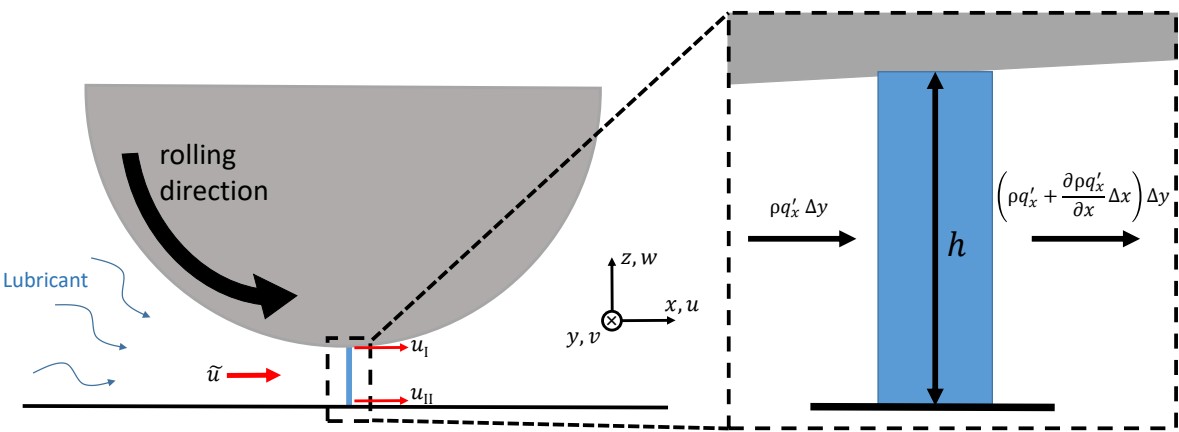

**Figure 1.** A control volume of fluid lying between moving bearing surfaces with surface tangential velocities $u_\text{I}$ and $u_\text{II}$.

Hydrodynamic lubrication is generally associated with conformal surfaces (e.g. journal bearings) and negligible elastic deformations. Boundary lubrication occurs when surfaces are no longer separated by a lubricant film and there is significant surface-surface contact. Mixed lubrication represents an intermediate state in which some penetration of the lubricant film has occurred, such that the load is shared between asperity contacts and fluid pressures. For non-conformal contacts fully elastohydrodynamic lubrication is aspired to, with mixed and boundary cases representing increasing levels of friction and a heightened risk of wear related damage.

## 3    Reynolds equation and the elastohydrodynamic lubrication problem

Lubricated conjunctions can support applied loads as a result of pressure distributions generated through fluid film interactions. The differential equation governing these interactions is known as the Reynolds equation. While derivations and applications of this equation are commonplace, a proper discussion of nuances occurring in the current problem requires a more detailed understanding of the equation's origin and underlying terms. As such, key elements from the derivation based on the laws of viscous flow and mass conservation will be presented, for more detailed considerations see Hamrock et al. (2004) and Dowson (1962). The full EHL problem is then defined. Note, a table of symbols is provided in Appendix A. The derivation of Reynolds equation begins by considering the rectangular control volume shown in Figure 1 of height $h$, width $\Delta x$ and length (into the page) $\Delta y$. The mass of lubricant contained within this control volume at any point in time is $\rho h \Delta x \Delta y$, where $\rho$ is the lubricant density. The rate at which this mass changes over time is determined by the difference between mass flowing into and out of the control volume. From Figure 1, mass flow differences in $x$- and $y$-directions are given by,

$$-\frac{\partial \rho q_x'}{\partial x}\Delta x \Delta y \ \text{ and } \ -\frac{\partial \rho q_y'}{\partial y}\Delta y \Delta x, \tag{2}$$

where,

$$q'_x = \int_o^h u \, dz \quad \text{and} \quad q'_y = \int_o^h v \, dz, \tag{3}$$

are volume flow rates per unit length/width. Conservation of mass requires that the rate at which mass is accumulated in the control volume equals the total difference between mass flowing into and out of the volume. Thus,

$$-\frac{\partial \rho q'_x}{\partial x} \Delta x \Delta y - \frac{\partial \rho q'_y}{\partial y} \Delta y \Delta x = \frac{\partial}{\partial t} \left( \rho h \Delta x \Delta y \right), \tag{4}$$

from which $\Delta$ terms cancel such that,

$$-\frac{\partial \rho q'_x}{\partial x} - \frac{\partial \rho q'_y}{\partial y} = \frac{\partial}{\partial t} \left( \rho h \right). \tag{5}$$

Applying zero-slip boundary conditions at lubricant-solid interfaces, and performing a number of integrations[3], it can be shown that volume flow rate expressions take the form (Hamrock et al., 2004),

$$q'_x = \underbrace{-\frac{h^3}{12\eta} \frac{\partial p}{\partial x}}_{\text{Poiseuille flow}} + \underbrace{\frac{u_{\mathrm{I}} + u_{\mathrm{II}}}{2} h}_{\text{Couette flow}} \tag{6}$$

$$q'_y = -\frac{h^3}{12\eta} \frac{\partial p}{\partial y} + \frac{v_{\mathrm{I}} + v_{\mathrm{II}}}{2} h. \tag{7}$$

$\eta$ is the lubricant dynamic viscosity and $p$ is pressure. As indicated, these flow rates contain Poiseuille and Couette contributions. Poiseuille flow is that driven by pressure gradients in the fluid, whereas Couette flow is induced by surface velocities $u_{\mathrm{I}}$ and $u_{\mathrm{II}}$ (shown in Figure 1); more specifically by shear stresses resulting from a viscous fluid interacting with moving boundary surfaces.

Substituting Equations 6 and 7 into Equation 5 and defining the mean entrainment velocities,

$$\tilde{u} = \frac{u_{\mathrm{I}} + u_{\mathrm{II}}}{2}, \quad \tilde{v} = \frac{v_{\mathrm{I}} + v_{\mathrm{II}}}{2}, \tag{8}$$

the general Reynolds equation, which governs the pressure distribution in fluid film lubrication, is obtained;

$$\frac{\partial}{\partial x} \left( \frac{\rho h^3}{12\eta} \frac{\partial p}{\partial x} \right) + \frac{\partial}{\partial y} \left( \frac{\rho h^3}{12\eta} \frac{\partial p}{\partial y} \right) = \frac{\partial}{\partial x} \left( \rho h \tilde{u} \right) + \frac{\partial}{\partial y} \left( \rho h \tilde{v} \right) + \frac{\partial}{\partial t} \left( \rho h \right). \tag{9}$$

In a bearing context, the full EHL problem consists of finding a solution to the Reynolds equation which also satisfies the following conditions of *total film thickness* and *load balance*,

$$h = h_m + h_g + h_e, \tag{10}$$

$$\iint p(x,y,t) \, dxdy = w(t). \tag{11}$$

---

[3]For example, this includes integrations: $\frac{\partial}{\partial z} \left( \eta \frac{\partial u}{\partial z} \right) \to \frac{\partial u}{\partial z} \to u \to \int u$

Equation 10 stipulates that, at each location, total film thickness ($h$) must equal the sum of surface separation components resulting from: minimum separation ($h_m$), un-deformed surface geometry ($h_g$) and local elastic deformations ($h_e$) caused by resulting pressures in the system. Equation 11 stipulates that the pressure distribution must balance the applied force, $w(t)$, at each point in time. Evaluation of elastic deformations is discussed further in Section 3.1, below.

As presented, the Reynolds equation is valid for variable-viscosity compressible flows, with changes in viscosity and density driven primarily by pressure (under isothermal conditions). Variations in lubricant properties are usually captured via empirical equations; for example, the *Barus law* and *Roelands equation* (Hamrock et al., 2004),

$$\eta = \eta_0 e^{\alpha p}, \quad \eta = \eta_0 10^{-[1.2 + \log_{10}(\eta_0)]\left[1 - (1 + p/2000)^{Z_1}\right]}, \tag{12}$$

(respectively) are both commonly used under isothermal conditions to characterise changes in viscosity. $\eta_0$ is the lubricant dynamic viscosity at the inlet temperature and for (gauge-pressure) $p = 0$, $\alpha$ is the pressure-viscosity coefficient and $Z_1$ a dimensionless pressure-viscosity index. Viscosity also varies strongly with temperature. In practise, viscosity information is normally provided at two reference temperatures, with interpolation allowing an appropriate value for $\eta_0$ (at the inlet temperature) to be identified (ASTM, 2020). For non-isothermal computational EHL modelling, empirical equations along the lines of Equation 12 that also include temperature have been developed (Hamrock et al., 2004). Similar equations are used to describe density variations with pressure, for example,

$$\rho = \rho_0 \left( 1 + \frac{0.6p}{1 + 1.7p} \right), \tag{13}$$

for some mineral oils, in which a roughly linear initial increase in density with pressure levels off to a maximum of $+35.3\%$ as $p \to \infty$. In Equation 13, different coefficients may be used depending on the lubricant. More accurate representations have also been developed which rely on greater numbers of coefficients. Empirical equations of the above types are developed within a specific range of conditions, hence, the validity of applied empirical relationships should be considered when looking to solve any given EHL problem. It should be noted that the Barus law, while easily implemented and useful for gaining an intuitive understanding of pressure-viscosity effects, is known to provide a poor approximation of real lubricant viscosity variations with pressure. Indeed, there is a growing call for more realistic modelling of lubricant rheological behaviour in general (Bair et al., 2016; Bair, 2019). It is argued that these aspects of EHL must be properly accounted for before it can be considered a truly quantitative discipline (Vergne and Bair, 2014; Bair et al., 2016; Bair, 2019). Despite these shortcomings, there still remain many examples where numerical models employing the above empirical equations are able to accurately recreate results obtained experimentally (e.g. see Tsuha and Cavalca (2020); Venner and Wijnant (2005); Zhang et al. (2020)).

## 3.1 Approximations in EHL modelling

The outlined derivation and EHL problem definition together provide an accessible justification of Equations 9-11, however, it should be noted that approximations are present for which proper consideration requires a first-principles derivation, and associated discussions, starting from the Navier-Stokes and continuity equations. Similarly, elastic deformations of bearing surfaces are usually resolved by approximating each body as an elastic half-space. As with any model, it is important to

understand the approximations being made and the conditions under which they are valid. Therefore, these aspects of numerical EHL models will be briefly outlined.

First, the fluid context is considered. EHL of machine components generally results in film thickness values which are small compared to the length of the conjunction through which the lubricant is flowing (see Section 6, below). Denoting typical surface separation in the conjunction by $h_0$ and typical conjunction length (over which changes in separation occur) by $L$, in the limit $h_0/L \to 0$ the inertial terms in the Navier-Stokes momentum equations disappear and one also obtains $\partial p/\partial z = 0$, *i.e.* pressure becomes constant across the film. Volume flow-rate expressions (Equations 6 and 7) are then obtained

via integration of the simplified, and now quasi-steady, momentum equations. Reynolds equation (Equation 9) follows by applying an integrated form of the continuity equation, ensuring mass-flow conservation. The approximation, $h_o/L \approx 0$, is known as the "lubrication approximation" and is valid in cases where $h_o/L \ll 1$. See Panton (2013) and Hamrock et al. (2004) for more details. Note, the derivation outlined in Section 3 implicitly uses this same approximation.

The problem of two curved elastic bodies in contact can be reduced to that of a single "equivalent" elastic ellipsoid or

140 cylinder contacting a rigid plane (Dowson, 1967; Harris and Kotzalas, 2006b). The geometry of the equivalent ellipsoid or cylinder is captured by the reduced radii of curvature $R_x$ and $R_y$ in $x$ and $y$ directions (see Appendix A). With respect to conjunction shape and deformation, undeformed surface geometries are generally approximated as being parabolic, resulting in,

$$h_g = \frac{x^2}{2R_x} + \frac{y^2}{2R_y}. \tag{14}$$

Local deformations are most commonly evaluated by treating each body as an elastic half-space, to which is applied the same pressure distribution as exists in the lubricated conjunction. This simplification allows deformations to be evaluated using relatively simple integral formulas (Johnson, 1987; Evans and Hughes, 2000). The magnitude of deflections resulting from an applied distribution of pressure is governed by the contacting materials' elastic moduli ($E$) and Poisson ratios ($\nu$), which may be combined into a single reduced modulus of elasticity, $E'$, for the equivalent elastic ellipsoid or cylinder (see Appendix

150 A). The half-space approximation is valid only if: surface geometries close to the contact region roughly approximate a plane surface, strains within the contact region are small enough to be evaluated using linear elasticity theory, and, stress fields resulting from pressures in the conjunction are not strongly influenced by body boundaries. These requirements are satisfied if the significant dimensions of the contact region are small with respect to the dimensions of the contacting bodies and the relative radii of curvature of the surfaces (Johnson, 1987). The same conditions on dimensions and curvature also ensure validity of

155 the parabolic geometries approximation. Hertzian contact theory, which concerns the (dry) contact of non-conforming elastic solids, is closely related and relevant to the lubrication problem (e.g. see Section 4). Hertzian theory assumes contacting surfaces (no longer separated by a lubricant layer) are frictionless, that the contact patch is elliptical or rectangular, and applies the same approximations as outlined above for the evaluation of geometries and deflections (Johnson, 1987). While the EHL problem requires numerical modelling to solve, Hertzian analysis of dry contact yields elegant analytical formulae describing

load-deflection relationships.

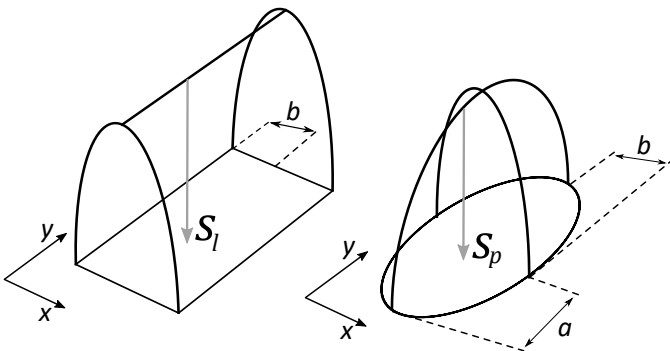

**Figure 2.** Surface normal-stress distributions in line (left) and point (right) contacts.

## 4 Line and point contacts

The current section deals with instances of dry contact. When an elastic solid is acted on by a load, deformation will occur. Cases of two contacting and loaded solids result in the formation of a contact patch, with the geometry of contacting solids determining the shape of the contact patch. Components which initially contact along a line (e.g. a cylindrical roller contacting a raceway), referred to as *line contacts*, lead to a rectangular contact patch with a semi-cylindrical surface normal-stress distribution which remains identical along its length[4] (Harris and Kotzalas, 2006b), see Figure 2. For an applied load per unit length of roller, $w_l$, this takes the form,

$$s_l(x) = \frac{2w_l}{\pi b} \left( 1 - \left( \frac{x}{b} \right)^2 \right)^{1/2},$$ (15)

where $b$ is the contact width. Line contacts can be considered as the limiting case of a long (elliptical) point contact, see below, or the problem can be reduced to that of line-loading on a two-dimensional elastic half-space (Johnson, 1987). Components which initially contact at a single point (e.g. a ball or spherical roller bearing contacting a raceway), referred to as *point contacts*, lead to an elliptical contact patch and semi-ellipsoidal surface normal-stress distribution (Harris and Kotzalas, 2006b),

$$s_p(x,y) = \frac{3w}{2\pi ab} \left( 1 - \left( \frac{x}{b} \right)^2 - \left( \frac{y}{a} \right)^2 \right)^{1/2},$$ (16)

for $w$ the applied load and with maximum normal-stress located at the contact centre. $a$ and $b$ are the elliptical contact dimensions, see Figure 2. With respect to pressures acting within a contact patch note that, under Hertzian contact, the pressure distribution being applied across each surface (by the other) must equal the surface normal-stress distribution which results (Equations 15 and 16). Therefore, Equations 15 and 16 also describe the pressure distributions acting within the contacts, commonly referred to as the "Hertzian pressure distributions".

Contact patch geometry is captured by the *ellipticity* parameter, this being the ratio,

$$k = \frac{a}{b},$$ (17)

---

[4]Ignoring end-effects and roller crowning.

of the elliptical semi-major and semi-minor axes $a$ and $b$, respectively[5]. For a given Hertzian point contact, as load is applied the axes expand proportionately to each other. Hence, $k$ remains constant and is a function of undeformed surface geometries only. The value of $k$ in a particular contact case is determined by an implicit equation, involving elliptic integrals, which requires iterative solving. Approximate formulas have been developed to allow for fast evaluation of the ellipticity parameter and associated elliptic integrals (Brewe and Hamrock, 1977; Hamrock and Dowson, 1977a; Antoine et al., 2006). The EHL problem outlined above applies equally to both contact types, with the line contact case often simplified to a 2D axis-symmetric problem in which side-leakage ($y$, $v$ and $\partial/\partial y$ terms) is neglected and the applied load at time $t$, $w(t)$, becomes the applied load per unit length of roller, $w_l(t)$.

Given simplifications associated with line contact EHL, it has proved useful in some analyses to consider the concept of an *equivalent line contact* representation of a point contact. Taking a point contact with applied load $w$ and patch dimensions $a$ and $b$, a 2D line contact representation is sought which shares its rolling direction patch width ($b$), geometry ($R_x$) and centreline ($y = 0$) stress distribution under dry contact. These conditions can be shown to hold for the distributed load $w_l = 3w/4a$ applied to a line contact whose $x$-direction geometry matches that of the point contact, but, which has the adjusted reduced modulus $\tilde{E}' = \left(1 + \frac{R_x}{R_y}\right)\mathcal{E}^{-1}E'$. $\mathcal{E}$ is an elliptic integral of the second kind whose value depends on $k$ (*e.g.* see Hart (2020)). Full details of this equivalent line contact formulation are provided in Appendix B. Other approaches to these types of equivalence have been taken in the literature; for example, seeking an equivalent line contact in which the maximum or mean Hertzian pressure coincides with that of the point contact for cases where patch widths ($b$) in the line and point contact don't coincide (Hamrock and Dowson, 1981).

## 5    Full EHL solutions

The first complete solution to an EHL problem was presented by Dowson and Higginson in 1959 (Dowson and Higginson, 1959) for four line contact cases. All calculations were carried out by hand, using mathematical tables and mechanical calculators (Hooke, 2009). Much important work followed this initial breakthrough, but it wasn't until almost two decades later that computing power became sufficient to allow EHL solutions in the point contact case to be obtained (Lubrecht et al., 2009). Since then, a plethora of significant advances have followed regarding numerical solvers for EHL problems, including: development of advanced multilevel multigrid solvers (Venner, 1994; Venner and Lubrecht, 2000a, b); full coupling of elastic and hydrodynamic equations - the *differential deflection method* (Evans and Hughes, 2000; Hughes et al., 2000; Holmes et al., 2003) - which enhances algorithmic efficiency and stability; and computational fluid dynamics implementations. These listed solvers apply the elastic half-space approximation for evaluation of deflections. A *full system approach* has also been developed which incorporates full-body elasticity using finite element methods (Habchi et al., 2008; Lugt and Morales-Espejel, 2011; Habchi, 2018). Implementation of EHL in multibody dynamics software modelling has also been considered (Dlugoš

---

[5]Note, no single convention holds for the allocation of axis labels ($x/y$) and contact patch dimensions ($a/b$). In the current work, $x$ is taken to be the direction of rolling and $a$ the semi-major axis of the contact patch, with line contacts treated as long elliptical contacts in this context. For the types of rollers/contacts seen in wind turbine main-bearings (*i.e.* where the semi-major axis of contact lies transverse to the direction of rolling) this allocation results in the normal-stress distributions shown in Equations 15 and 16.

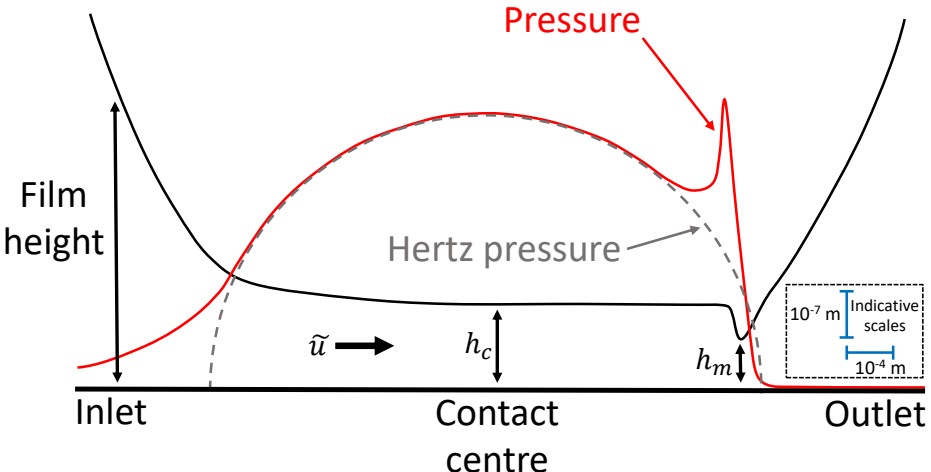

**Figure 3.** General characteristics of EHL contacts, including indicative orders of magnitude for vertical and horizontal scales. Also shown are the central and minimum film thickness values, $h_c$ and $h_m$ respectively. Note, the depicted features would be expected to occur within a narrow central section of the conjunction shown in Figure 1.

and Novotnỳ, 2021). For an overview of recent developments in numerical EHL modelling see (Meng et al., 2020). Accurate EHL solutions are now generated routinely and fairly easily even for complex cases such as those involving time-varying loads and speed, moving surface roughness or mixed lubrication conditions. However, it should be noted that this only holds where solver code and relevant expertise are available, since setting up such solvers is highly non-trivial.

Simplified formulations of the EHL problem have also been developed which allow for analytical and semi-analytical solutions to be obtained (Wolveridge et al., 1970; Morales-Espejel and Wemekamp, 2008; Greenwood, 1972). Such formulations, while approximate, are highly efficient and provide important insights into EHL behaviour and conditions, even proving useful when implementing full numerical solvers (e.g. they can help with the identification of appropriate mesh dimensions, as well as supporting interpretation and sense-checking of results).

## 6    General characteristics of EHL contacts

Figure 3 details characteristic features which tend to be present in EHL contacts[6], as has been confirmed extensively using numerical modelling and experimental investigations (Hamrock et al., 2004; Spikes, 2006; Albahrani et al., 2016; Foord et al., 1969; Wymer and Cameron, 1974). Significant elastic deformation can be seen to have taken place, resulting in a near parallel channel throughout most of the contact conjunction. Pressure at the inlet can be seen to rise rapidly to meet the Hertzian

(dry-contact) pressure curve (see Section 4), which is then tracked through much of the conjunction. Extremely high pressures develop within the contact gap, resulting in dramatic increases in lubricant viscosity and so dominance of the shear (Couette) driven terms of Equations 6 and 7. Prior to the outlet, a constriction occurs in the oil film, immediately after a sudden spike

---

[6]More specifically, Figure 3 shows a 'slice' through the contact in the direction of rolling, $x$.

in pressure. These features are coupled, with the pressure spike driven by the abrupt reduction in film height. The constriction itself is a consequence of mass-flow continuity as follows.

We briefly consider the simplified case in which flow in the $y$-direction (side-leakage) is ignored; focusing on the non-transient case, $\partial(\rho h)/\partial t = 0$, mass-flow continuity (Equation 5) requires,

$$\rho q_x' = \text{constant}, \tag{18}$$

throughout. Considering Equation 6 at different points in the conjunction: at the entrance $\partial p/\partial x > 0$ and so the Poisuille term will act against the Couette flow, in the centre of the contact it has already been indicated that Poisuielle flow is minimal and Couette flow dominant (so mass flow $\sim \rho \tilde{u} h$); however, in the exit region, decreasing pressures ($\partial p/\partial x < 0$) lead to Poiseuille and Couette terms acting in the same direction while, simultaneously, rapid reductions in viscosity are taking place - increasing the Poiseuille term magnitude. From Equations 6 and 18 it is clear that in order to avoid flow discontinuity, a reduction in $h$ or $\rho$ (or both) must occur.

In practise it has been found that a marked reduction[7] in film thickness occurs close to the outlet (as shown in Figure 3) in both incompressible and compressible cases. This is true both with and without side-leakage. In the latter case the pressure spike magnitude is dramatically reduced relative to incompressible results (one example in Hamrock et al. (2004) sees a reduction of 3.7 times). The central film thickness, $h_c$, is smaller (on the order of tens of %) for compressible flow under otherwise identical conditions, while the minimum film thickness, $h_m$, only changes by a few percent (Venner and Bos, 1994; Hamrock et al., 2004).

For line contacts the features shown in Figure 3 are present along much of the roller length, with distortions known to occur at roller ends (Wymer and Cameron, 1974). For point contacts the same features are present, but arranged in a *horseshoe* which tracks the elliptical contact patch boundary. Central film thickness, representative of much of the conjunction, is still at the contact centre; while minimum film thickness tends to occur at two side-lobes, away from the centreline (Foord et al., 1969). Graphical depictions of typical film thickness variations across point and line contact conjunctions are shown in Figure 4. The horseshoe and side-lobe characteristics of point contacts arise due to the presence of Poiseuille-term driven lateral flow (side-leakage). In such cases, mass-flow continuity[8] again indicates the presence of a constricted band aligned perpendicular to conjunction outflow, but, with outflow velocities now vector values containing $u$ and $v$ components (not to be confused with $\tilde{u}$ and $\tilde{v}$, see Figure 1). The perspective of relative (Poiseuille vs Couette) flow contributions, and implications for $u$ and $v$ values at different points, is important when interpreting effects of load, speed and ellipticity on described contact features. An excellent account of such analysis may be found in Wheeler et al. (2016a).

Additional relevant characteristics of EHL contacts (both point and line) are as follows:

1. Both minimum, $h_m$, and central, $h_c$, film thickness values are important for understating conditions within the contact conjunction. The former, combined with surface roughness information, indicates the degree to which separation of

---

[7]$(h_c - h_m)/h_c \times 100\%$ values of between 17 and 70% have been reported in the literature (Chaomleffel et al., 2007), with operating conditions being a strong driver.

[8]Following a similar argument to that outlined above.

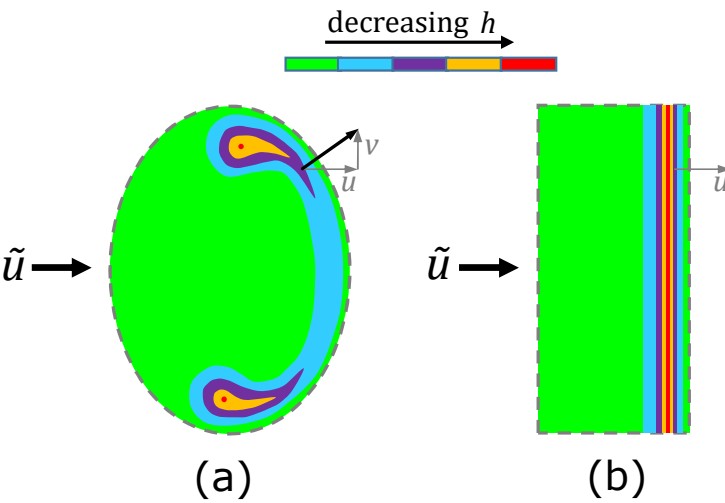

**Figure 4.** Typical film thickness variations in (a) point and (b) line contact conjunctions, end effects (see Appendix C) are not shown in the latter case. Note, these depictions are purely conceptual, film height values in one should not be interpreted as necessarily being equal to those in the other.

surface asperities has been achieved; while the latter allows good representation of traction/friction conditions throughout most of the almost parallel gap.

2. With respect to operating conditions, the entrainment velocity ($\tilde{u}$) is known to be the main driver of lubricant film thickness. The relative effect of load is significantly smaller, attributable to the fact that load changes coincide with an expansion or contraction of the contact patch. Material properties are not insignificant, but in practise only a narrow range of values will apply in any given rolling bearing situation. The lubricant viscosity at the inlet, $\eta_0$, also plays an important role in determining the resulting film thickness.

3. As load increases and/or entrainment velocity decreases, surface geometries and pressures converge to those of dry Hertzian contact. The pressure spike also reduces such that maximum pressure occurs at the contact centre and equals that of dry contact.

4. As the ellipticity of point contact geometry increases[9], the elliptical conjunction (Figure 4a) tends asymptotically to a line contact conjunction (Figure 4b). This effect can be understood in the context of relative flow, since $v \to 0$ as ellipticity increases.

---

[9]Elongating Figure 4a vertically.

## 7 Dimensionless groupings and film thickness equations

When modelling a physical system via a set of equations, such as for EHL, it is possible to re-express the problem in an equivalent dimensionless form which generally depends on a reduced number of, also dimensionless, parameters. These dimensionless parameters are constructed as appropriate products, powers and ratios of dimensional quantities appearing in the original set of equations. In reduced and dimensionless form the problem is simplified and generalised, with effects of interacting physical phenomena elucidated. To illustrate this last point, consider a system for which solutions depends on parameters $q_1$ and $q_2$ with physical units in common. It may be the case that system response (e.g. flow rate, wave height etc.) is proportionately increased by $q_1$, but decreased by $q_2$. In such a scenario, ultimate response is driven by the dimensionless quantity $q_1/q_2$, rather than each value independently. The relevant number of parameters to characterise response is therefore reduced, and the interaction between effects associated with $q_1$ and $q_2$ made clear. *Dimensional analysis* or *similarity analysis* are the names given to the study and application of such ideas and associated methods.

In EHL, the following parameters comprise the most common set of dimensionless groupings used to describe lubrication conditions in line contact conjunctions (Dowson, 1967; Harris and Kotzalas, 2006a):

$$W_l = \frac{w_l}{E' R_x} \qquad \text{(load)} \tag{19}$$

$$U = \frac{\eta_0 \tilde{u}}{E' R_x} \qquad \text{(speed)} \tag{20}$$

$$G = \alpha^* E' \qquad \text{(material).} \tag{21}$$

Recall that $w_l$ is load per unit length. $\alpha^*$ is the lubricants inverse asymptotic isoviscous pressure coefficient, $1/\alpha^* = \int_0^\infty \frac{\eta_0}{\eta} dp$, a quantity which can be directly determined for a given lubricant using high-pressure viscometer measurements.

---

$\underline{\alpha^* \text{ vs } \alpha}$

The pressure-viscosity coefficient $\alpha$, generally obtained via curve-fitting to measurements, has commonly been used instead of $\alpha^*$ for EHL analyses. Indeed, for a lubricant that really did follow a Barus law pressure-viscosity curve (see Equation 12) it is easily shown that $\alpha^* = \alpha$, with the substitution therefore appropriate in this case. However, in practise the Barus law does not provide a good characterisation of viscosity variations with pressure. The coefficient $\alpha^*$, first proposed by Blok in 1963, characterises a lubricants pressure-viscosity behaviour as a single parameter, but without assuming any particular functional form for $\eta(p)$. Other candidate coefficients have also been proposed in the literature, but, for the purposes of estimating EHL film thickness (using existing formulas) it has been demonstrated that the inverse asymptotic isoviscous pressure coefficient, $\alpha^*$, is the one which should be used (Vergne and Bair, 2014). All such coefficients are temperature dependent, with the $\alpha^*$ used in film thickness equations necessarily being the value corresponding to the lubricant inlet temperature, $T$. For further discussion of these aspects of EHL modelling, see Bair (1993); Vergne and Bair (2014).

---

In the case of point contacts, speed and material parameters remain unchanged whereas dimensionless load becomes (Dowson, 1967; Harris and Kotzalas, 2006a),

$$W = \frac{w}{E'R_x^2},\tag{22}$$

where $w$ is the applied load. Point contact geometry is captured in dimensionless form by the ellipticity parameter, $k$, or equivalently by $D = R_y/R_x$. Equivalence follows from the fact that $k$ can be expressed as a function of $D$ only (Masjedi and Khonsari, 2015a). Non-dimensional film thickness is given by $H = h/R_x$.

Dimensionless groupings are commonly identified using a combination of intuition, experience and trial-and-error. However, systematic processes exist by which a minimal, or "optimal", set of dimensionless quantities can be identified that fully determine system behaviour as defined by governing equations (Moes, 1992; Hsiao, 2001). It should be emphasised that identification processes, and the resulting optimal dimensionless sets, depend on the governing equations and boundary conditions of a problem only. Note, also, that minimal sets of dimensionless quantities for a given problem tend not to be unique. While the number of elements in each minimal set will be the same, alternative choices for the groupings of dimensional variables are generally present. All alternative groupings which form a minimal set can be identified if required (Hsiao, 2001).

For the EHL problem (as defined by equations in Section 3) in line and points contacts, optimal parameter analysis reduces the three parameters above (load, speed and material) to just two (Moes, 1992; Hsiao, 2001), a load parameter,

$$M_l = w_l \left(\frac{1}{\eta_0 \tilde{u} E' R_x}\right)^{1/2} = W_l (2U)^{-1/2} \quad \text{(line contact)} \tag{23}$$

$$M = w \left(\frac{1}{\eta_0^3 \tilde{u}^3 E' R_x^5}\right)^{1/4} = W (2U)^{-3/4} \quad \text{(elliptical contact)}, \tag{24}$$

and viscosity parameter,

$$L = \alpha^* \left(\frac{\tilde{u}\eta_0 E'^3}{R_x}\right)^{1/4} = G(2U)^{1/4}. \tag{25}$$

The factors of 2 are present to coincide with Moes (1992), in which entrainment speed is taken to be the sum, rather than mean, of surface velocities. While this is the form generally used (Marian et al., 2020; Wheeler et al., 2016b), the parameters are equally valid with the factors of 2 removed. Both forms have appeared in the literature (Moes, 1992; Hsiao, 2001), hence, it is important to ascertain which has been applied if comparing operating conditions or applying related film thickness equations. Non-dimensional film thickness in the optimal parameter case takes the form $H = (h/R_x)(2U)^{-1/2}$. The same analysis identifies $D^{-1}$, introduced above, as the parameter representing contact patch geometry in the elliptical case (Hsiao, 2001). Additional parameters are required to fully capture more complex viscosity and density characteristics, for full details see Hsiao (2001). Despite the proven reduction in the number of parameters required to characterize EHL conditions, use of $W_l$ (or $W$), $U$ and $G$ persists, although film thickness equations utilising reduced sets of parameters have been developed (Marian et al., 2020). The parameters $M_l$ (or $M$) and $L$ are, however, commonly used when plotting operating regions and results, since visualisation becomes clearer and easier with a reduced number of variables.

Having identified a set of dimensionless parameters (optimal or otherwise) which determine the response of the system defined by governing equations, it follows that features of interest (e.g. $h_m$ and $h_c$) will also be determined by these same

parameters. If such relationships can be sufficiently well approximated by analytical equations, fast evaluation and analysis of key features becomes possible without requiring complex numerical solvers to be implemented in every case. Film thickness formulas have therefore been developed by performing least-squares curve fits between outputs of full EHL solvers and analytical expressions containing dimensionless parameters. Equations 26 and 27 present two of the earlier equations identified this way for estimating minimum film thickness ($h_m$) in line and point contacts, respectively,

**Line contacts** (Dowson, 1967):

$$\frac{h_m}{R_x} = 2.65 \frac{U^{0.70} \, G^{0.54}}{W_l^{0.13}} \tag{26}$$

**Point contacts** (Hamrock and Dowson, 1977a):

$$\frac{h_m}{R_x} = 3.63 \frac{U^{0.68} \, G^{0.49}}{W^{0.073}} \left(1 - e^{-0.68k}\right) \tag{27}$$

Despite the early stage at which they were developed, these equations provide remarkably accurate estimates and are still used today (Harris and Kotzalas, 2006a). Equation 26 is based on Barus law viscosity modelling and Equation 27 on Roelands equation viscosity modelling. The relative importance of speed, load, viscosity and material properties (as described at the end of Section 6) may be seen to be reflected in the exponent values of the above equations. Quite a number of subsequent refinements have been undertaken using larger datasets generated by more advanced solvers (Marian et al., 2020). Some of the most extensive fitting was undertaken for line and point contacts in Masjedi and Khonsari (2012) and Masjedi and Khonsari (2015a), respectively, resulting in the following equations:

**Line contacts** (Masjedi and Khonsari, 2012):

$$\frac{h_m}{R_x} = 1.652 \frac{U^{0.716} G^{0.695}}{W_l^{0.077}} \tag{28}$$

**Point contacts** (Masjedi and Khonsari, 2015a):

$$\frac{h_m}{R_x} = 1.637 \frac{U^{0.711k^{-0.023}} \, G^{0.65k^{-0.045}}}{W^{0.09k^{-0.15}}} \left(1 - 0.974e^{-0.676k}\right) \tag{29}$$

Both of these equations are based on Roelands equation viscosity modelling. Unfortunately, the full range of dimensionless parameter values over which these formulas were fitted appears to have been misrepresented in the literature. In Wheeler et al. (2016b), and then reproduced in Marian et al. (2020), parameter limits for the point contact $h_m$ equation are given (approximately) as $15 \leq M \leq 10^4$ and $5 \leq L \leq 20$. While these limits are those of the cases shown in the results tables of Masjedi and Khonsari (2015a), it is explicitly stated that only a subset of the full analysis is reproduced there. Moreover, the full range of dimensionless parameter values used for curve fitting are also given. Taking the stated limiting values in Masjedi and Khonsari (2015a), the domain across which these equations were fitted is in fact bounded by $2.82 \leq M \leq 1.47 \times 10^5$ and $2.97 \leq L \leq 28.20$. The true domain of validity for these equations is therefore significantly larger than has been reported. For completeness, parameter limits for the line contact equation (Masjedi and Khonsari, 2012) are also stated; limits in $L$ match those of the point contact case and for $M_l$, $2 \leq M_l \leq 353.55$. With respect to ellipticity, Equation 29 was developed across the

range $1 \leq k \leq 8$. At $k = 8$, film thickness predictions match closely those of the line contact formula (Equation 28) applied to an "equivalent line contact" representation of the conjunction (see Section 4). It was therefore recommended that the equivalent line contact approach is taken for cases where $k > 8$ (Masjedi and Khonsari, 2015a).

The same studies which result in formulas for $h_m$ also use curve-fitting to identify similar formulas for $h_c$ (Hamrock and Dowson, 1977a; Masjedi and Khonsari, 2012, 2015a; Marian et al., 2020).

### 7.1 Accuracy of film thickness equations

It is important to appreciate that analytical film thickness equations provide estimated values based on curve fitting within a specific range of dimensionless parameter values, *i.e.* operating conditions. Applicability limits for any given equation must therefore be checked and respected for each case being analysed. Furthermore, isothermal conditions and Newtonian fluid behaviour are also often assumed. Correction factors for such effects have been proposed (Marian et al., 2020), these are subject to the same limitations as outlined for the film thickness equations themselves. In the context of curve-fitting, derived formulas are able to recreate the numerical results on which they are based to a high degree of accuracy. For example, comparisons between Equation 29 and numerical fitting data result in a mean error of 3.27% and a maximum error of 9.79% (Masjedi and Khonsari, 2015a); note, it is not clear whether these numbers relate to the full dataset or only a subset. Similarly, for the line contact $h_m$ equation (Equation 28) a maximum error of 10.41% is reported (Masjedi and Khonsari, 2012). This is certainly promising, but, investigating accuracy at points not included in the fitting set is crucial to forming a full picture of equation performance. In Wheeler et al. (2016b) a range of point contact analytical film thickness equations for $h_m$ and $h_c$ were compared in this way. Maximum observed errors across tested $h_m$ equations occurred for the circular contact case ($k = 1$), reaching about 90%[10]. In general, $h_m$ was found to be overestimated by analytical equations, with the severity of over-estimation increasing as load increases or entrainment speed decreases. When ellipticity was increased to $k = 2.92$, errors in $h_m$ predictions reduced to satisfactory levels (about 6% on average). The study concluded that Equation 29 (along with its companion $h_c$ equation) should be used for cases of long elliptical contacts ($k > 1$). Other equations are recommended for circular contacts ($k = 1$), while the slender contact case ($k < 1$) remains an open problem (Wheeler et al., 2016b). The major conclusion of Wheeler et al. (2016b) is that current analytical equations must be considered as providing qualitative, rather than truly quantitative, estimates of film thickness. Note, consistent with the discussion concerning viscosity coefficients (above), Wheeler et al. (2016b) use the inverse asymptotic isoviscous pressure coefficient, $\alpha^*$, throughout their analysis.

Despite the above caveats, film thickness equations in many cases do provide good estimates of film values, in particular when lubricant properties are well known. $h$ values are generally better predicted in line contacts and long elliptical contacts and $h_c$ tends to be better predicted than $h_m$ (Chaomleffel et al., 2007; Wheeler et al., 2016b). As outlined above, $h_m$ tends to be over-predicted rather than under-predicted by analytical equations.

---

[10]Recent work (Habchi and Vergne, 2021) has confirmed that the presented film thickness equations struggle at predicting $h_m$ in point contacts ($k = 1$). An improved analytical approach is also presented therein.

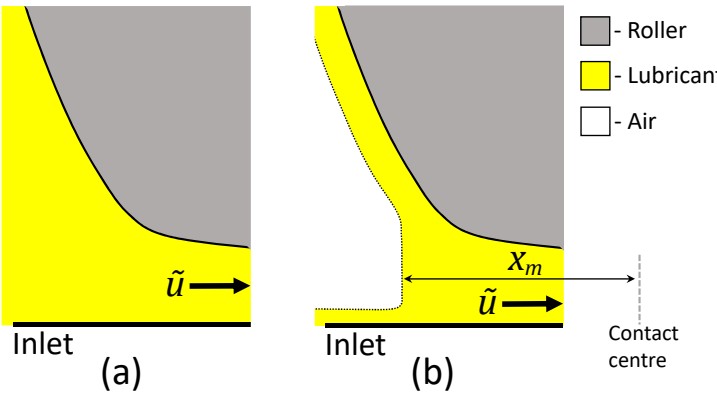

**Figure 5.** Graphical depictions of a lubricated contact inlet under (a) fully-flooded and (b) starved conditions.

## 8    Starvation

EHL behaviour and film thickness equations discussed above assume fully-flooded conditions, in which an adequate supply
of lubricant is available to the contact. Starvation is the term given to cases in which lubricant supply is insufficient. Figure 5
illustrates these different regimes. Under starved conditions, surface-to-surface filling of the inlet with lubricant only occurs
close to the contact edge. This reduces the magnitude of developed hydrodynamic pressures, in turn reducing the load carrying
capacity relative to fully-flooded conditions and, hence, the film thickness at any given load is also reduced. Starvation levels
are commonly characterised by the dimensionless inlet meniscus distance, $\tilde{m} = x_m/b$, where $x_m$ is shown in Figure 5. In
the case of line contacts, film thickness reduction formulas under starvation have been presented in the literature based on a
semi-analytical analysis of the starved EHL problem (Wolveridge et al., 1970) and curve-fitting to the results of numerical
integration of simplified lubrication equations (Goksem and Hargreaves, 1978). Starvation was studied in these works by
varying the inlet location in hydrodynamic pressure integrals. Note, the non-dimensional inlet distance used in these earlier
works differs from $\tilde{m}$, and the viscosity-pressure relationship was characterised using a Barus law. More recently, Masjedi
and Khonsari (2015b) undertook an extensive computational study of film thickness reductions under starvation, with surface
roughness also present. Similar to earlier work, starvation is generated by moving the inlet of the solver domain towards the
contact centre. A Roelands pressure-viscosity relationship was used. While in other work starvation levels are indicated by an
appropriately non-dimensionalised inlet distance, Masjedi and Khonsari instead propose that starvation be linked to the mass
flow rate through the starved contact ($\dot{m}_\mathrm{s}$) relative to fully-flooded conditions ($\dot{m}_\mathrm{ff}$). Their *starvation degree* is therefore defined
as the fractional reduction in flow rate,

$$\zeta = 1 - \frac{\dot{m}_\mathrm{s}}{\dot{m}_\mathrm{ff}}. \tag{30}$$

$\zeta$ takes values between 0 and 1, with 0 indicating fully-flooded conditions and 1 complete starvation. This definition has an
intuitive appeal, since film thickness is fundamentally linked to the quantity of lubricant moving through the contact. The
appropriateness of $\zeta$ in this context is evidenced by the quality of fits and simplicity of parametric equations obtained from

curve-fitting to model outputs. In the case of line contacts, $h_c$ reductions were found to be linear in $\zeta$, with $h_m$ reductions similar but weakly nonlinear. The latter fit takes the form (Masjedi and Khonsari, 2015b),

$$\frac{h_{m,s}}{h_m} = 1 - \zeta^{1.08}, \tag{31}$$

for which the maximum reported error across fitted point was less than 5%. This analysis closely mirrored those authors' previous work in which Equations 28 and 29 were presented.

In the case of point contacts, perhaps the most important earlier work was that of Hamrock and Dowson (Hamrock and Dowson, 1977b), who studied the effects of starvation on elliptical contact conjunctions computationally. As above, this was achieved by adjusting the inlet distance of their solver domain and using a Roelands pressure-viscosity equation. Based on a parametric study and subsequent curve-fitting, they proposed formulas for the reduction in central and minimum film thickness values under starvation, with the level of starvation indicated by $\tilde{m}$. For $h_m$ this takes the form,

$$\frac{h_{m,s}}{h_m} = \begin{cases} \left(\frac{\tilde{m}-1}{m^*-1}\right)^{0.25} & : \text{if } \tilde{m} < m^* \text{ (starved)} \\ 1 & : \text{if } m \geq m^* \text{ (fully-flooded),} \end{cases} \tag{32}$$

where,

$$m^* = 1 + 3.34 \left(\frac{R_x h_m}{b^2}\right)^{0.56}. \tag{33}$$

$m^*$ represents the transition point to starved lubrication, this being the dimensionless inlet distance at which the minimum film thickness begins to change as $\tilde{m}$ is reduced further. $h_m$ is the fully-flooded minimum film thickness, and $h_{m,s}$ the minimum film thickness under starvation. While a valuable contribution, important aspects of starved flow regimes in point contacts were not included, as will be outlined below.

**Lubricant flow characteristics under starvation:** Figure 6 shows characteristics features, from above, of fully-flooded and starved EHL in line (subfigures a and b) and point (subfigures c and d) contacts. In line contacts, ignoring end effects (see Appendix C), side leakage is negligible and so little or no lubricant is displaced laterally. It follows that in this case it is reasonable to treat the quantity of lubricant (and so the meniscus distance) available at each point along the contact as being the same, even in a full bearing in which the lubricant supply to each roller is influenced by the passage of previous rollers. In such line contact cases a straight meniscus of equal height will be present, conforming to starvation as nominally modeled by moving the inlet towards the contact centre. In point contacts, things are significantly different. In fully flooded conditions, Figure 6c, the lubricant remains enclosed about the contact, ensuring a sufficient supply into the conjunction. However, as the oil supply is reduced or speed/viscosity are increased, the flow regime transitions to the "butterfly" shape shown in Figure 6d. Flow speeds (including lateral flow contributions) interact with effects of surface tension and viscosity such that the lubricant film ruptures behind the contact, resulting in the bulk of the lubricant being displaced to the sides of the rolling track where separated sidebands form (Poll et al., 2019; Fischer et al., 2021b). In a lubricated point contact rolling bearing, sidebands at the outlet of one roller form the inlet to the next roller, explaining the inflow of Figure 6d. If no oil flowed from the

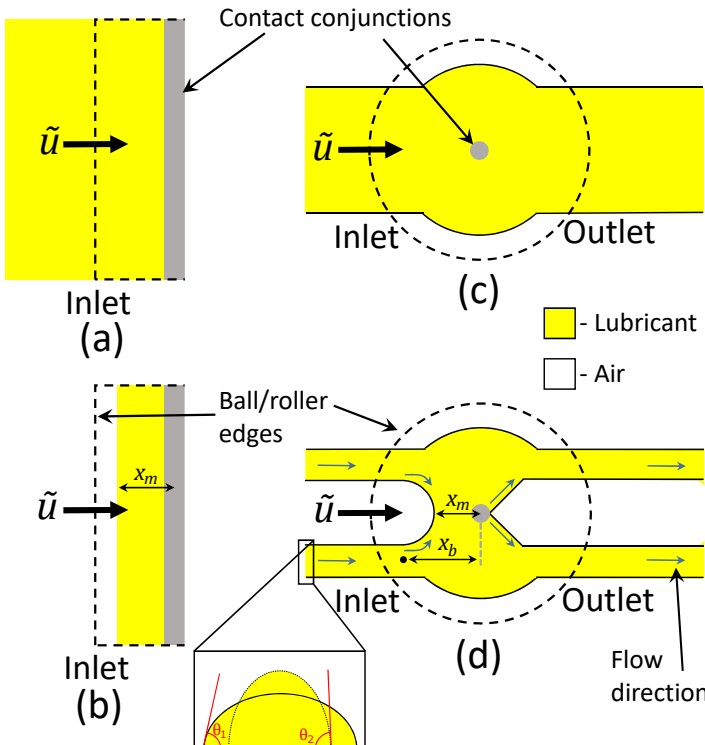

**Figure 6.** Graphical depictions of lubricant flow characteristics in fully-flooded and starved regimes for line (a-b) and point (c-d) contacts.

sides to the middle of the track, the small amount remaining across the track centre[11] after roller passage would rapidly and monotonically deplete during subsequent over-rollings, meaning a steady-state is never achieved. However, experimental evidence and bearing operational experience have both shown that, in general, a steady-state level of starvation is reached, indicating a balance between lubricant feed and loss mechanisms. Replenishment (also called reflow) of starved point contacts must therefore be taking place. Out-of-contact replenishment can occur, wherein the oil/air surface tension drives a flattening of lubricant sidebands between roller passes, causing lubricant to flow back towards the track centre (Guangteng and Spikes, 1996; Fischer et al., 2021a). But, this mechanism is relatively slow and so only significant at low over rolling frequencies or in cases of very high sidebands[12] (Guangteng and Spikes, 1996). At exceedingly small film thicknesses (on the order of 10s of nm) direct van der Waal and related interactions elicit a powerful lubricant spreading effect, the disjoining pressure, which slows reductions in film thickness caused by increasing starvation levels and prevents total film collapse (Guangteng and Spikes, 1996). Disjoining pressure is therefore another form of out-of-contact replenishment. In typical bearing applications, close-to-contact replenishment is the dominant mechanism of reflow (Fischer et al., 2021a). Close-to-contact replenishment

---

[11]In reality the regions marked as 'air' in the figure contain an oil/air mix.

[12]This only occurs in situations where a copious supply of lubricant is present, with starvation driven by high rolling speeds rather than low lubricant volumes (Guangteng and Spikes, 1996).

takes place as follows: sideband height means that surface-to-surface filling with lubricant initially occurs at the sidebands (referred to as the *sideband meniscus*, occuring a distance $x_b$ from the contact centre, as shown in Figure 6d). Immediately downstream of each sideband meniscus, lateral flows are induced which separate the flow volume such that part of the lubricant volume is drawn towards the track centre (also shown in Figure 6d), with the remainder displaced around the contact. This mechanism determines the amount of oil in front of the contact conjunction, and so also the meniscus distance, $x_m$. The larger $x_b$, the greater the amount of oil that flows to replenish the contact inlet, increasing $x_m$. Similarly, a reduction in $x_b$ reduces oil replenishment to the contact, decreasing $x_m$. Beyond the standard parameters which characterise EHL (see Section 7), the inlet meniscus distance under starvation is additionally effected by flow behaviour, wetting behaviour and the total volume of oil in the system (Fischer et al., 2021a, b). Lubricant flow behaviour is dictated by the relative magnitude of viscous forces to surface tension ($\sigma_{\text{oil,air}}$) forces, as captured by the capillary number,

$$C_a = \frac{\eta u}{\sigma_{\text{oil,air}}}. \tag{34}$$

Wetting behaviour relates to the effects of lubricant properties on fluid surface formation at solid/fluid/air boundaries. In the literature, this has been characterised using the contact angle, $\theta$, occurring at the three way interface (see Figure 6d) (Fischer et al., 2021a, b). The impact of these factors on $x_m$ has been explored in the literature and, as demonstrated in Fischer et al. (2021a), each may be understood in the context of the effect on $x_b$, as follows. $x_m$ has been shown to reduce as $C_a$ increases (Nogi et al., 2018; Fischer et al., 2021a). The associated increase in viscous force contributions (relative to surface tension) results in sidebands being pushed further out from the track centre. Due to the geometry of a point contact the vertical distance between track and roller is greater here, delaying sideband meniscus formation and so reducing $x_b$, which in turn reduces $x_m$. An increase in $\theta$ (as shown for $\theta_2 > \theta_1$ Figure 6d) or an increase in the total oil volume both result in higher lubricant sidebands. This causes the sideband meniscus to form earlier, increasing $x_b$ and so also $x_m$.

Returning to film thickness formulas under starvation, it should be clear that limitations are present for the outlined Hamrock & Dowson starvation analysis, in the context of an operating rolling bearing, which led to Equations 32 and 33. Specifically, a straight inlet meniscus was assumed with no provision for sidebands and other effects described above. Despite this, good agreement has been demonstrated between numerical and experimental results and the Hamrock & Dowson analytical starvation equations (Nogi, 2015a, b; Nogi et al., 2018) in some cases. This indicates that,

1. The dimensionless inlet distance, $\tilde{m}$, does indeed appear to be a key factor determining film thickness reductions under starvation

2. Equations 32 and 33 can provide good estimates of film reductions under starvation (in some instances) so long as $\tilde{m}$ and $m^*$ are known or can be well estimated.

$m^*$ requires the minimum film thickness under fully-flooded conditions, $h_m$, which may be estimated using equations described in Section 7. The same caveats to that discussion apply again here. A remaining piece of the puzzle is therefore being able to estimate the dimensionless inlet distance, $\tilde{m}$, for given operating conditions and contact geometry and while accounting for real world characteristics of starved flow. The discussion of starved flow characteristics, above, indicates that this might be

done through the adoption of an expanded parameter set for equation fitting. Such an approach has been undertaken by Nogi *et al.* (Nogi, 2015b; Nogi et al., 2018) for circular and elliptical point contacts. Starved flow was modelled using a numerical EHL solver which accounts for film rupture behind the contact as driven by $C_a$. Sideband formation was also included by considering conservation of mass, after film rupture, at the outlet of the computational domain. The outlet oil distribution was then fed in at the inlet to simulate repeated over-rollings while neglecting out-of-contact replenishment. Simulations were run until a steady-state was achieved. A Roelands pressure-viscosity relationship was used. Numerical model results were shown to agree well with experimental measurements (Nogi, 2015a, b; Nogi et al., 2018), with analytical formulas then fitted to the numerical results for a range of dimensionless parameter values. In Nogi (2015b) a circular point contact ($k = 1$) meniscus distance formula is presented which includes $C_a$ and $h_i$. The latter is the initial film height, uniform across the inlet, used when initiating the numerical simulations. $h_i$ relates to the total oil volume available, although not in the most straightforward manner. Nogi et al. (2018) then also includes ellipticity, $k$, when fitting equations. In each case, equations are adapted to account for a nonuniform inlet meniscus across the contact. In the notation of the current paper, the elliptical point contact meniscus distance equation (for $h_m$) presented in Nogi et al. (2018) is,

$$\tilde{m} = \left(\frac{h_i}{c_h}\right)^{0.5973+0.0217k} \left(\frac{h_{c,k=1}}{c_h}\right)^{-0.3437+0.249/k} \exp\left(-(0.708+0.792k)C_a^{0.63}\right)(1+0.19\exp(1-k)). \tag{35}$$

$c_h$ is the Hertzian approach of the contact, capturing its relative size and geometry. $h_{c,k=1}$ indicates the fully-flooded central film thickness when $k = 1$. It should be noted that axis selection in the original paper is such that the ellipticity ratio appearing there is the inverse of $k$ as defined in this paper. The circular contact specific equation of Nogi (2015b) is fitted over a larger range of parameters and so may provide better results for that case. More recently, Fischer et al. (2021b) applies the CFD model presented in Fischer et al. (2021a) to identify a formula for $\tilde{m}$ which is dependent on $C_a$, $\theta$ and the total available oil volume. This model uses CFD simulation to study oil flow in the vicinity of point contact geometry. Modelled roller and raceway surfaces are rigid and the gap height set manually, remaining fixed throughout. Curve fitting to outputs of a parametric study resulted in the meniscus distance equation (Fischer et al., 2021b),

$$\tilde{m} = 4.533 \exp\left(-2.571 C_a^{0.384}\right) \cos(\theta)^{-0.249} \left(V_{l,\text{oil}} \times 10^{-6}\right)^{0.446} b^{-1}, \tag{36}$$

where $V_{l,\text{oil}}$ is the available oil volume per unit length of track in $\mu l/mm$. This equation relates to the case of a circular point contact. It is interesting to note the similarities in Equations 35 and 36 with respect to the terms and exponents involving $C_a$ and information on the available oil volume. Equation 36 does not contain a fully-flooded film thickness term. Fischer et al. (2021b) also compares experimentally measured central film thickness values under starvation with those obtained applying Equations 36 and the central film thickness equivalent of Equation 32. $h_c$ values, as speed (and so $C_a$) is increased, predicted by the analytical equations were found to fall much more precipitously than the measured values. Furthermore, measured film thickness values leveled off at 80-90nm, behaviour reminiscent of disjoining pressure effects described in Guangteng and Spikes (1996), although occurring here at higher film thickness values. At present it is unclear what is causing this disparity. It could be that in tested cases Equation 36 fails (for whatever reason) to provide a good estimate of the real meniscus distance, or that Equation 32 is unable to accurately characterise starvation here, even if $\tilde{m}$ is well estimated. Whether or not one of

these is true depends on the physical effects present and whether they are captured in the models used to develop predictive equations. Based on similarities between the two $\tilde{m}$ expressions, the Nogi *et al.* equations would not be expected to fare much better at predicting the behaviour seen experimentally in Fischer et al. (2021b). Despite disparities in predicted and measured film reductions under starvation, the results in Fischer et al. (2021a) indicate that Equation 36 appears to capture the point of starvation onset (*i.e.* where $\tilde{m} = m^*$) to a reasonable degree of accuracy, and so might be applicable in assessing whether a given bearing in specified conditions is expected to operate in the fully-flooded or starved regime. However, the utility of the presented equation in this regard has only been shown for a small number of experimental cases and so further work would would be needed to determine if this holds in general. Since reasonable agreements with experimental data have been seen for starvation predictions in other work (Nogi, 2015a; Nogi et al., 2018), a more comprehensive comparison of experimental data with numerical model and analytical equation predictions is required in order to 1) ascertain under which conditions current equations (*i.e* Equations 32, 33, 35 and 36 among others) are viable 2) identify the physical processes leading to disparities between measured and predicted film reductions 3) seek to improve/extend analytical starvation equations such that identified additional effects are accounted for. Starvation as charactised by $\zeta$ (Masjedi and Khonsari, 2015b), rather than $\tilde{m}$, may also be worth further consideration. Note, a starved film reduction formula for point contacts is also presented in Masjedi and Khonsari (2015b), although it was developed under the assumption of a straight inlet meniscus. Beyond the required developments outlined above, a further issue which remains to be tackled is that of predicting starvation in a full bearing. While progress has been made on predicting $\tilde{m}$ values, current formulas require information on the total available oil volume. In Equation 31 this is captured by $\zeta$, and in Equations 35 and 36 this is captured by $h_i$ and $V_{l,\text{oil}}$, respectively. It is not yet clear how these values might be determined in practice for real bearings, especially where they have been operating for some time.

Since direct prediction of film thickness reductions under starvation in a full bearing is not yet possible in a practical sense, it is helpful to consider the typical order-of-magnitude effect of starvation on $h$ values. Film reduction magnitudes seen in the literature tend to be on the order of 10s of % (Harris and Kotzalas, 2006a; Masjedi and Khonsari, 2015b; Nogi et al., 2018). With some earlier literature suggesting it be assumed minimum film thickness values reduce to 71% of their fully flooded values (Hamrock and Dowson, 1981; Lugt, 2009), reported as 70% in Harris and Kotzalas (2006a), where starvation is suspected but starvation levels are unknown. These numbers should, however, be treated as crude order-of-magnitude estimates only since, in reality, film reductions due to starvation depend on a range of effects and operating parameters (as outlined above). Starvation is known to commonly occur in grease lubricated roller bearings, hence, starvation will be revisited in this context in Section 11.

## 9  Non-steady effects in EHL

Many EHL contacts operate under non-steady conditions in which load, speed and even contact curvature (the latter being the case during gear meshing) change with time. The critical timescale, $t_{\text{c}}$, determining the impact of such variations is the time it takes for a particle of lubricant to pass through the contact (Venner and Wijnant, 2005). When the time taken for conditions to vary is large, compared to $t_{\text{c}}$, non-steady effects become negligible and a quasi-static analysis is sufficient for characterising

film variations over time. For example, this is generally the case for roller bearings operating under constant load, in which each roller sees a continuous variation in applied force as it traverses the loaded zone. In the absence of additional effects, the film thickness variations are well captured by applying steady-state equations (Section 7) at each time step. However, if variations occur rapidly, meaning over timescales similar to or shorter than $t_{\text{c}}$, local surface deformations are induced at the inlet which then propagate through the contact. Non-steady EHL phenomena have been studied numerically, including via semi-analytical models, and experimentally (Hooke, 2003; Venner and Wijnant, 2005; Morales-Espejel, 2008; Zhang et al., 2020). Steady-state formulae do not capture variations in film thickness and pressure within the contact resulting from rapid non-steady effects, their use in such cases therefore risks over-estimating minimum film thickness values (Hooke, 2003). Piezoviscous behaviour plays an important role in non-steady EHL. Recall, as discussed in Section 6, that very high viscosities and Couette flow dominance hold in the central region of a loaded EHL contact. As the lubricant film passes into the central region, it therefore becomes very "stiff" and moves through the conjunction, at approximately $\tilde{u}$, as a shear flow that is independent (for the most part) of the contact inlet. In the case that entrainment speed is suddenly increased, increased film thicknesses occur at the contact inlet (via local surface deformations) which are then carried through the conjunction at the new entrainment speed (Hooke, 2003). The new, increased, film thickness values are only seen across the whole contact once the original values have passed out of the conjunction. While the increased entrainment speed ($\tilde{u}$) is seen simultaneously across the whole contact, a greater film thickness requires an increased volume of lubricant in the conjunction. For a shear dominant flow, this can only happen through the advective process described above, irrespective of pressure changes at the inlet. This explains why film changes from rapid speed adjustments aren't uniform across the contact. While in steady-state conditions film thickness sensitivity to load is known to be small, the effects of rapid load variations can be dramatic (Venner and Wijnant, 2005; Hooke, 2003). As load increases, the width ($b$) of the contact also increases, meaning the contact edge is moving rapidly in the opposite direction to entrainment. The result is an augmented "effective" entrainment speed,

$$\tilde{u}_{\text{eff}} = \tilde{u} + \frac{db}{dt}, \tag{37}$$

for the inlet. Due to the stiff nature of the central film, this results in the formation of a dome-shaped entrapment of lubricant at the inlet which is then carried through the conjunction. Similarly, a rapid reduction in load results in negative values of $db/dt$, and so a reduced effective entrainment speed at the inlet. In this case, a local drop in film thickness forms at the inlet and moves through the contact (Venner and Wijnant, 2005). A conceptual representation of the latter case is provided in Figure 7. Somewhat counter intuitively, it is therefore the case that rapid increases in load can temporarily increase the minimum film thickness, while rapid load reductions can temporarily decrease the minimum film thickness. Periodic load variations have also been considered in the literature. Film height variations through the contact are accompanied by local variations in pressure and material stress (Hooke, 2003). For time-varying loads, the rate of expansion or contraction, $db/dt$, relative to entrainment speed, $\tilde{u}$, has been proposed as a criteria for determining whether a quasi-static analysis may be used (Hooke, 2003). Non-steady effects were found to become important where,

$$\frac{|db/dt|}{\tilde{u}} \, (\times 100\%) \geq 25\%. \tag{38}$$

While effective entrainment speed plays an important role in non-steady EHL, squeeze film effects ($\partial(\rho h)/\partial t$ in Equation 9) also strongly influence the resulting film thickness variations over time (Zhang et al., 2020). Both effects must therefore be accounted for.

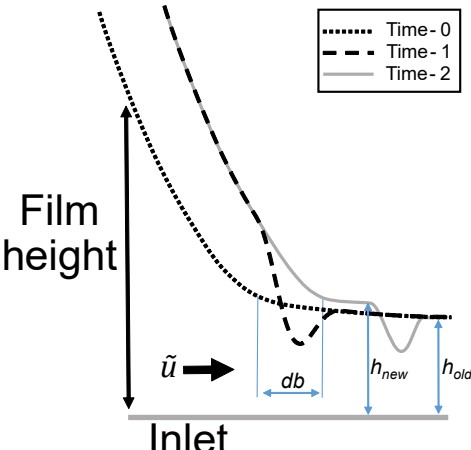

**Figure 7.** Conceptual depiction of the effect of a rapid load reduction for an EHL contact. Steady-state conditions hold initially (Time - 0), before a rapid load reduction causes the contact width to reduce (Time - 1). Movement of the contact edge causes a reduction in effective entrainment speed of $db/dt$, reducing hydrodynamic pressures within the inlet. Because of this, the larger material deformations close to the contact edge cannot be sustained. Hence, a local reduction in film thickness has formed (Time - 1) which then passes through the stiff central region of the contact mostly unchanged (Time - 2). The contact interior is initially unaffected, only altering as the local perturbation passes through and new steady-state film heights enter the conjunction.

EHL contacts also experience non-steady conditions due to intermittent operation, *i.e.* starts and stops along with associated periods of acceleration/deceleration. Piezoviscous behaviour again plays an important role in such cases (Sugimura et al., 1999). Lubricant entrained into a contact at start-up initially forms a "front" which passes through the conjunction at constant height before characteristic features of the EHL contact (see Section 6) are then established (Glovnea and Spikes, 2001). Secondary fronts have also been found to occur in some cases (Glovnea and Spikes, 2001), giving the initial film a stepped profile. Under cases of very high acceleration, oscillatory film thickness behaviour has been observed (Glovnea and Spikes, 2001) which has been linked, in part, to dynamics of the overall mechanical system within which an EHL contact is operating (Popovici et al., 2004). In instances of shutdown, halting of operation sees an EHL oil film begin to collapse, initially in a uniform manner (Ohno and Yamada, 2007). As the deceleration reduces the entrainment speed further, a local minima forms at the contact inlet. Together with the minima at the contact outlet, these features form an entrapment of lubricant within the contact. Subsequent reductions of the film thickness within the entrapment are slow; in some cases entrapped films remain almost unchanged for hours or days (Ohno and Yamada, 2007). The thickness of the initial entrapped lubricant film has been shown to increase with increasing values of the parameter,

$$\alpha\eta_0 \times deceleration, \tag{39}$$

indicating that both rheological and operational effects are important (Ohno and Yamada, 2007). Time variations in surface-oil-layer thickness distributions also occur within lubricated components. Such oil migration is driven by gravitational and surface tension effects after operation is halted (Gao et al., 2022), and (additionally) by centrifugal forces while operating (Van Zoelen et al., 2008, 2010).

## 10  Surface roughness interactions

As discussed in Section 2, film thickness must be considered relative to surface roughness in order for the lubrication regime to be known. However, it is also the case that rough surface micro-geometry will interact with the lubricant flow and deform elastically, with both effects influencing surface separation and lubrication conditions. Much work has been undertaken over the years in what is now known as micro-EHL, leading to significant advances in understanding and modelling capabilities. An excellent overview is provided by Morales-Espejel (2014). The presence of roughness results in part of the load being carried by surface asperities (Masjedi and Khonsari, 2015a), as opposed to being carried purely hydrodynamically. Such interactions are important when considering micropitting of machine elements (Morales-Espejel, 2014). With respect to conditions in the lubricated conjunction, roughness has been shown to result in both "mean" and "local effects". Mean effects are overall modifications to surface separation and pressure, relative to an equivalent smooth contact. Local effects are in the form of local variations in film height and pressure which move through the contact and, hence, are non-steady in nature. The mean effect resulting from the presence of homogeneous surface roughness is an increase in film thickness, but, by an amount that is smaller than the change in surface $\sigma$ (Morales-Espejel, 2014; Masjedi and Khonsari, 2015a). Therefore when roughness increases, $h_m$ increases, but $\Lambda$ decreases. Note, more structured roughness can have a different effect (Morales-Espejel, 2014). Film thickness equations presented in Section 7 are those for smooth surfaces. Additional multiplicative factors have been identified, also via curve fitting, which account for surface roughness effects. For the Masjedi and Khonsari line and point contact minimum film thickness equations these, respectively, take the form (Masjedi and Khonsari, 2015a, 2012):

$$\Phi_l \quad = \quad 1 + 0.026 \left(\frac{s_{\text{std}}}{R_x}\right)^{1.120} V^{0.185} W_l^{-0.312} U^{-0.809} G^{-0.977} \tag{40}$$

$$\Phi_p \quad = \quad 1 + 0.141 \left(\frac{s_{\text{std}}}{R_x}\right)^{1.073} V^{0.149} W^{-0.044} U^{-0.828} G^{-0.954} k^{-0.395}, \tag{41}$$

where $s_{\text{std}}$ is the standard deviation of surface heights (assuming normally distributed roughness, $s_{\text{std}} = \sigma$) and $V = v_h/E'$ is a dimensionless hardness number, with $v_h$ the surface Vickers hardness. From analysis across standard ranges of operating parameters, line and point contact results for dimensionless roughness levels, $s_{\text{std}}/R_x$, of around $1 \times 10^{-6} - 5 \times 10^{-6}$ or less have been shown to be well approximated by smooth surface results (Masjedi and Khonsari, 2015a, 2012). Note, the above modification factors may be applied where $\Lambda > 0.5$, but are no longer valid if the film parameter falls below this value (Masjedi and Khonsari, 2015a, 2012).

## 11 Grease lubrication

The vast majority of rolling element bearings are grease lubricated, where "grease" may be defined as a dispersion of a thickening agent in a liquid lubricant (Lugt, 2009). Lubricant base oil is held inside sponge-like structures of thickener fibres through a combination of Van der Waal and capillary forces. The resulting semi-solid consistency is beneficial due to its ease of use, good sealing action and corrosion resistance. However, this same consistency generally leads to starved lubrication conditions (Poll et al., 2019), since grease will not reflow (at a macroscopic level) back to the rolling track after being swept out by the passage of rolling elements. The total quantity of grease directly participating in the separation of contacting surfaces is therefore reduced. The initial phase of grease redistribution, the "churning phase", occurs within the first ∼10 hours of operation after a bearing has been freshly charged with grease (Cen and Lugt, 2020). Once this initial grease flow has ceased, the bearing enters the "bleeding phase" in which swept grease reservoirs are generally only able supply lubricant to the contacts by releasing ("bleeding") oil through phase separation (Lugt, 2016). Oil is also mechanically released from the thickener network by over-rolling, principally in the churning phase (Lugt, 2009). The starved lubrication contribution of the bled-oil portion of a grease lubricated contact may be treated as described in Section 8 (Fischer et al., 2021a). However, other effects can also be present, as will be outlined. Understanding, modelling and predicting grease lubrication is difficult. This is because thickener and base oil interactions result in nonlinear shear stress - shear rate behaviour, even at low shear rates and pressures. The apparent viscosity of grease also decreases continuously over time while being sheared, then recovers once shearing stops (thixotropy) (Lugt, 2009, 2013). Over longer timescales, grease properties change as the thickener structure deteriorates due to being mechanically worked (Cen et al., 2014). Oxidation also slowly degrades grease performance. Due to these complexities, there is as yet no complete theory which allows film thickness in grease lubricated bearings to be accurately and consistently predicted in general. While a complete theory of grease lubrication is not yet established, significant advances have been made regarding the key mechanisms and interactions at work. A summary of pertinent results in this field will therefore be outlined:

1) *The thickener contributes to film thickness at low speeds*: At higher speeds the film thickness observed in fully-flooded grease lubricated contacts often coincides with that of oil lubrication using the grease base-oil (Morales-Espejel et al., 2014; Cen et al., 2014; Kanazawa et al., 2017; Nogi et al., 2020). However, this is not always the case since the bulk grease and bled-oil can have significantly different rhealogical properties to the base oil, even at high speeds (Cousseau et al., 2012). Still, it is common at higher speeds for fully-flooded grease lubrication to be strongly determined by the viscosity of the base oil. As speed is reduced, film thickness initially reduces inline with the behaviour predicted by standard film thickness equations (See Section 7), but, eventually a "transition speed" is reached after which further reductions in speed result in increasing film thickness values. The rate of increase in this region, as speed is decreased, can be similar to that seen for oil lubrication as speed is increased, meaning these low speed grease effects are significant with respect to resulting film thicknesses. Speed versus film thickness plots for fully-flooded grease lubrication therefore exhibit a characteristic 'V' shape (Cen et al., 2014; Morales-Espejel et al., 2014; Kanazawa et al., 2017). The described behaviour at low speeds results from entrained thickener fibers becoming the dominant driver of surface separation, further evidenced by the fact that in this region the film thickness, for a given grease, is independent of base oil viscosity and temperature (Kanazawa et al., 2017). As the grease is mechanically

worked over time, the thickener structure degrades and constituent particles become smaller. This reduces film thickness values seen in the low speed region, whereas the higher speed region is unaffected (Cen et al., 2014). The degradation process has been found to primarily occur within the first 100h of operation. The transition from low-speed to higher-speed behaviour has been found to be dependent on film thickness (Kanazawa et al., 2017). More specifically, experimental findings indicate that the described "low-speed" effects occur for film thicknesses $h < c\mathcal{D}$, for some constant $c$ and where $\mathcal{D}$ is the diameter of thickener fibres or possibly that of entangled fibre networks[13] (Kanazawa et al., 2017). Recent modelling work (Nogi et al., 2020), validated against experimental results, has elucidated the possible mechanisms at play. In Nogi et al. (2020), fully-flooded grease lubrication of a point contact is modelled by characterising the base-oil as a Newtonian fluid, and the thickener network as a porous plastic medium which is also a non-Newtonian fluid. Their findings indicate that the thickener concentration remains at that of the bulk grease throughout the conjunction when speeds are high. However, at low speeds the base oil becomes more easily squeezed out of the inlet than the thickener network itself[14], resulting in significant increases in thickener concentration (both within and around the contact) and order-of-magnitude increases in the film thickness, relative to the base oil alone. The increase in thickener concentration leads to an increase in the equivalent viscosity of lubricant entering the conjunction. Considering typical EHL behaviour (see Sections 7), increased film heights in this setting would therefore be expected. Increases in $h_m$ for grease lubrication at low speeds may be understood in the context of reduced side-leakage as a result of the equivalent viscosity increase within the conjunction (Nogi et al., 2020; Lugt, 2009), see Section 6. The concept of an equivalent/effective viscosity for lubricating grease has been applied previously. For example, in Morales-Espejel et al. (2014) it is demonstrated that effective-viscosity variations (with speed), for a range of greases, can be well described by analytical expressions containing just two free parameters. Identifying parameter values which characterise a particular grease requires (at a minimum) only two measured datapoints of speed and film thickness, at the same temperature. Extrapolation to other temperatures may then be performed using standard viscosity-temperature equations (Morales-Espejel et al., 2014; ASTM, 2020). Despite its relative simplicity, this approach was shown to perform well when comparing predicted film thickness values with experimental data across a range of temperatures. It is emphasised that this method, and indeed all of the phenomena described in this first item, relate specifically to conditions of fully-flooded grease lubrication.

2) *Starved grease lubrication is dependent on bled oil availability and replenishment, as well as thickener deposits on contact surfaces*: As indicated towards the beginning of this section, the hydrodynamic component of starved grease lubrication is as described in Section 8, with bled-oil providing the liquid lubricant to line and point contacts. The available volume and properties of bled-oil from the applied grease therefore determine the rate of contact replenishment, and so the hydrodynamic contribution to film thickness. Additionally, it has been shown that over-rolled and broken-down thickener fibers can deposit on the contacting surfaces, forming a thin layer (Cann et al., 1992; Cann, 1996; Hurley and Cann, 1999; Poll et al., 2019). The total film thickness under starved grease lubrication is therefore the sum of hydrodynamic and deposited layer components (Poll et al., 2019; Fischer et al., 2021a). Furthermore, the wettability properties[15] of oil on a thickener layer can be significantly

---

[13]This is our interpretation of Kanazawa et al. (2017) results, summarising their findings and proposed mechanisms of grease film formation.

[14]Due to differing behavioural changes exhibited by the base oil and thickener network under a decreasing shear rate (Nogi et al., 2020).

[15]The contact angle, $\theta$, in the context of Section 8.

different to those of oil on steel (Huang et al., 2016). Therefore, thickener deposits also directly influence the hydrodynamic film component.

3) *Grease lubrication is fundamentally non-steady*: It has been observed experimentally that, even after the churning phase has ended, fluctuations in temperature for grease lubricated bearings occur which are irregular and of varying duration (Lugt, 2013). This is true even under constant operating conditions. Temperature time-series may also look very different to each other for identical bearings run under the same conditions. Some such instances of temperature fluctuation may be caused by later cases of churning, due to a grease lump breaking away and entering the raceway. However, further experiments (Lugt, 2013) demonstrated that they most commonly occur as part of a repeating cycle of,

i) starvation driven film breakdown and metal-to-metal contact, leading to an increase in temperature

ii) lubricant replenishment and an increase in film-thickness, leading to a reduction in temperature

Replenishment resulting from an increase in temperature may be due to the softening and release of fresh grease, increased bleed rates and/or the increased mobility of bled-oil (*i.e.* a reduction in its viscosity). In addition to the above, a recent experimental study considered non-steady effects in grease lubricated starved contacts, showing that behaviour similar to that outlined in Section 9 can also be observed for grease lubrication (Zhang and Glovnea, 2020).

4) *Close-to-contact replenishment has been shown to be the dominant reflow mechanism in grease lubricated full bearings*: Recent work has explored ball-bearing contact replenishment in real bearings (Cen and Lugt, 2019, 2020). For bearings with different numbers of balls (hence different timescales between contact passes) and different cage geometries, the normalised film thickness (relative to fully-flooded conditions) was found to be a function of the parameter *speed × viscosity × half contact-width*. Since these quantities are local to the contact and independent of the time between ball passes, it may be concluded that contact replenishment is a local phenomenon in these grease lubricated bearings. Also consistent with the results discussed in Section 8, it was shown that increases in speed and/or viscosity lead to increased levels of starvation (Cen and Lugt, 2019, 2020) and reduced film thickness values (Cen and Lugt, 2020).

As previously stated, understanding of grease lubrication has advanced significantly. It should be noted that much of the recent progress, outlined here, has been developed for point contacts either as single contacts or within ball-bearing test rigs (although some important work has also been undertaken for roller bearings containing line contacts (Lugt, 2009, 2013)). The relative importance of described effects may therefore differ in practise, depending on the roller type, the characteristics of applied loading and the distribution of load within the bearing *etc*. General film thickness formulas for grease lubrication are not yet available. As stated, there is a consensus that grease lubricated bearings are usually operating under starved conditions. Similar to the oil lubrication case, grease starvation effects on film thickness are usually of the order of 10s of % (Cen and Lugt, 2019, 2020), with one study proposing starved grease film thickness values be estimated as 70% of the fully flooded value (under oil lubrication), assuming a viscosity equal to that of the base-oil (Lugt, 2009). This coincides with the film reduction levels discussed at the end of Section 8. However, as previously, this should be treated as a crude order-of-magnitude estimate only.

## 12 Conclusions

EHL theory has come an exceptionally long way since its inception around 1950. Incredibly complex elastohydrodynamic lubrication problems are now solved routinely, with experimental comparisons demonstrating the effectiveness of currently available models. However, important further work remains, much of which is centred around the need for more realistic modelling of lubricant rheological behaviour and the extension of results from single contacts to full bearings. Therefore, elastohydrodynamic lubrication is (and will likely remain) a complex and rapidly evolving field. The current review has attempted to ensure the details of these complexities are accessible to a more general engineering audience, in order to support cross-disciplinary understanding with respect to this field and future interdisciplinary work. It is again emphasised that when applying film thickness equations, or indeed any other output of lubrication modelling, the approximations, assumptions and overall validity of applied equations should be considered on a case by case basis, allowing predictions to be properly contextualised. Such considerations should also dictate which equations are used. The theory presented in this review is applied in "Part 2" of the study in order to consider lubrication in a wind turbine main-bearing.

# Appendix A: Table of symbols

$a$ — semi-major contact dimension, assumed here to lie transverse $(y)$ to the rolling direction (m)

$b$ — semi-minor contact dimension, assumed here to lie in $(x)$ the rolling direction (m)

$E_\mathrm{I}, E_\mathrm{II}$ — Youngs modulii of solids I and II (Pa)

$E'$ — reduced modulus of elasticity (Pa), $2/E' = (1 - \nu_\mathrm{I}^2)/E_\mathrm{I} + (1 - \nu_\mathrm{II}^2)/E_\mathrm{II}$

$G$ — dimensionless material parameter (-), $G = \alpha^* E'$

$h, h_m, h_c$ — film thickness, minimum film thickness and central films thickness, respectively (m)

$k$ — ellipticity parameter (-), $k = a/b$

$L$ — dimensionless viscosity parameter (Moes) (-), $L = G(2U)^{1/4}$

$M_l, M$ — dimensionless load parameter (Moes) for line and point contacts respectively (-), $M_l = W_l(2U)^{-1/2}$, $M = W(2U)^{-3/4}$

$p$ — pressure (Pa)

$r_{Ix}, r_{IIy}, \ldots$ — radius of curvature of surface I/II in the $x/y$ direction, presented here as a strictly positive quantity (m)

$R_x$ — reduced radius of curvature in the entrainment direction (m), $\frac{1}{R_x} = \frac{sgn(\mathrm{I}x)}{r_{\mathrm{I}x}} + \frac{sgn(\mathrm{II}x)}{r_{\mathrm{II}x}}$

$R_y$ — reduced radius of curvature transverse to the entrainment direction (m), $\frac{1}{R_y} = \frac{sgn(\mathrm{I}y)}{r_{\mathrm{I}y}} + \frac{sgn(\mathrm{II}y)}{r_{\mathrm{II}y}}$

$sgn(\cdot)$ — $sgn(\mathrm{I}x)$ is 1 if surface I is convex in the $x$-direction and $-1$ if it is concave in the $x$-direction (similarly for II and/or $y$)

$s_l, s_p$ — surface stress in line and point contacts, respectively (Pa)

$s_\mathrm{std}$ — standard deviation of surface heights (assuming normally distributed roughness, $s_\mathrm{std} = \sigma$) (m)

$T$ — lubricant inlet temperature (°C)

$u_\mathrm{I}, u_\mathrm{II}$ — tangential velocities, in the entrainment direction $(x)$, of surfaces I and II at the contact location (m/s)

$\tilde{u}$ — mean entrainment velocity (m/s), $\tilde{u} = (u_\mathrm{I} + u_\mathrm{II})/2$

$U$ — dimensionless speed parameter (-), $U = \eta_0 \tilde{u}/(E' R_x)$

$v_\mathrm{I}, v_\mathrm{II}, \tilde{v}$ — similar to "$u$" terms, but transverse $(y)$ to entrainment direction (m/s)

$V$ — dimensionless hardness number, $V = v_h/E'$, for $v_h$ the surface Vickers hardness (-)

$w$ — normal load in point contact (N)

$w_l$ — normal load per unit length in line contact (N/m)

$W_l, W$ — dimensionless load parameter for line and point contacts respectively (-), $W_l = w_l/(E' R_x)$, $W = w/(E' R_x^2)$

$\alpha$ — pressure-viscosity coefficient of the lubricant (at the inlet temperature, $T$) (Pa$^{-1}$)

$\alpha^*$ — inverse asymptotic isoviscous pressure coefficient (at the inlet temperature, $T$) (Pa$^{-1}$), $1/\alpha^* = \int_0^\infty \frac{\eta_0}{\eta} dp$

$\Lambda$ — lubrication film parameter (-), $\Lambda = h_m/\sigma$

$\eta$ — lubricant dynamic viscosity (Pa $\cdot$ s)

$\eta_0$ — lubricant dynamic viscosity at the inlet temperature, $T$, and for (gauge-pressure) $p = 0$ (Pa $\cdot$ s)

$\nu_\mathrm{I}, \nu_\mathrm{II}$ — Poisson's ratios of solids I and II (-)

$\rho$ — lubricant density (kg $\cdot$ m$^{-3}$)

$\rho_0$ — lubricant density at the inlet temperature, $T$, and for (gauge-pressure) $p = 0$ (kg $\cdot$ m$^{-3}$)

$\sigma_\mathrm{I}, \sigma_\mathrm{II}$ — surface roughness, in the form of root-mean-square deviations, for surfaces I and II respectively (m)

$\sigma$ — combined roughness of contacting surfaces (m), $\sigma = \sqrt{\sigma_\mathrm{I}^2 + \sigma_\mathrm{II}^2}$

$\Phi_l, \Phi_p$ — film thickness modification factors accounting for surface roughness effects, line and point contacts respectively (-)

## Appendix B: Equivalent line contact formulation

Starting from Equations 15 and 16 and reducing $s_e$ to the case $y = 0$, the resulting stress distributions can only be identical if contact widths (b) are the same and if $w_l = 3w/4a$. Therefore, it only remains to ensure that the contact width for the equivalent line contact, under distributed load $w_l = 3w/4a$, matches that of the point contact under applied load $w$. Point contact semi-minor and semi-major axes, for the applied load $w$, take the following forms (Harris and Kotzalas, 2006b),

$$b = \left( \frac{6w\mathcal{E}}{\pi k \Sigma_\rho E'} \right)^{1/3}, \tag{B1}$$

$$a = \left( \frac{6k^2 w\mathcal{E}}{\pi \Sigma_\rho E'} \right)^{1/3}. \tag{B2}$$

$\Sigma_\rho = 1/R_x + 1/R_y$ is the point contact curvature sum. The line contact with equivalent $x$-direction geometry has curvature sum $\Sigma_{\rho,\text{line}} = 1/R_x$. The line contact with this geometry and reduced modulus $\tilde{E}'$ sees the following contact width, under distributed load $w_l$ (Harris and Kotzalas, 2006b),

$$b = \left( \frac{8w_l R_x}{\pi \tilde{E}'} \right)^{1/2}. \tag{B3}$$

Note, Equations B1-B3 all constitute standard formulae in Hertzian contact theory. Equating Equations B1 and B3, having substituted $w_l = 3w/4a$ and with $a$ given by Equation B2, it follows that $\tilde{E}' = \left( 1 + \frac{R_x}{R_y} \right) \mathcal{E}^{-1} E'$.

## Appendix C: Finite line contacts

End effects in finite length line contacts have not been considered in the current work. This is due to their relative complexity and specificity with respect to roller profiling. In general, end effects lead to the global minimum in film thickness actually occurring at the edges of line contact rollers. Roller profiling can mean that the global minimum is not too far from that along the roller centreline (the value predicted by line contact film thickness equations). Towards the centre of a finite roller it has also been shown that there are only small differences in the pressure and film thickness profiles when comparing with a semi-infinite 2D model. A key takeaway is that the centreline film thickness equations for line contacts may over-estimate the true value of $h_m$ across the roller. For more information see Mihailidis et al. (2013); Tsuha and Cavalca (2020) and further literature discussed therein.

*Author contributions.* This work was led by EH. All authors contributed to planning, writing and revising this paper.

*Competing interests.* The authors declare they have no competing interests.

*Acknowledgements.* This work forms part of Project AMBERS[16]. EH is funded by a Brunel Fellowship from the Royal Commission for the Exhibition of 1851. EdeM's PhD project is funded by the Powertrain Research Hub, co-funded by the Offshore Renewable Energy Catapult. The authors would like to acknowledge their help and support. Finally, the authors would like to thank the two anonymous reviewers for their helpful comments and suggestions.

---

[16] Advancing Main-BEaRing Science for wind and tidal turbines.

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
