# Peer review of "Wind turbine main-bearing lubrication - Part 1: An introductory review of elastohydrodynamic lubrication theory"

_Wind Energy Science, 2021_

## Referee Comment (RC1)

**Review: Wind turbine main-bearing lubrication - Part 1: An introductory review of elastohydrodynamic lubrication theory**

Hart et al.

The manuscript presents a review of Elastohydrodynamic (EHD) theory, starting from the very basic concepts to the application, assumptions and limitations of the theory. It is important to mention that this reviewer understands that the authors try to address the message to a community which might not be too familiar EHD. Thus the reviewer understands that some basic concepts need to stay in the present manuscript, otherwise a more synthetic manuscript could have been written.

The reviewer is favourable to the publication of the manuscript but requires some minor revisions or at least the answer of some questions.

Revisions:

1. Equation (12): There are increasing voices (Vergne and Bair 2014) arguing that these models are too simplistic to capture the real behaviour of viscosity as a function of pressure and temperature. This is not mentioned in this section although the topic is somehow mentioned in the discussion section.

2. Equations (26-27): The work of (Habchi and Vergne: https://doi.org/10.1007/s11249-021-01512-z) shows significant deviations among the different authors mentioned here and experimental results for minimum film thickness. Perhaps the authors should comment this.

3. In section: Grease Lubrication, two important references seem to be missing for the understanding of thickener contribution on the EHD film thickness: Nogi, (https://doi.org/10.1080/10402004.2020.1778147) and Morales-Espejel (Tribology International 74 (2014) 7–19), especially for slow rotating bearings.

4. In the Dynamic effects section, the reviewer remarks that starts-stops are not mentioned by the authors, in the targeted application they are multiple and very important, strictly speaking is not a dynamic effect but it is dynamic in nature. Some works have been written on EHL stopping or accelerating contacts with simple formulae.

---

## Referee Comment (RC2)

**Reviewer Blind Comments to Author**

The submitted paper gives an overview of the current state of elastohydrodynamic lubrication theory, focusing on simplified descriptions for reaching a wider readership. The authors are encouraged to add further figures in order to support the overarching goal of reaching a wider (non-expert) readership. The complex processes taking place in an EHL contact (e.g. starvation) can be illustrated using existing images. In the current version the reader is challenged to use his power of imagination. While the depth of the explanations is nicely balanced, the review leaves most of the chapters open. Please consider adding clearer recommendations for the aimed readership. Furthermore, multiple equations are given without proper citation. Therefore, the authors should assess if all references had been included. Furthermore, the title is misleading. The overview does not clearly explain the applicability or the relevance of the individual approaches for the case "main-bearing". In the reviewer's opinion the title should be changed or the relevance of each approach (including the validity range) for main-bearing applications should be addressed.

Please consider also following points:

**Introduction**

• Could the authors please give an overview of other EHL-Reviews (e.g. doi:10.3390/lubricants8050051) and shortly comment the differences between the reviews and the need for a new one?

**Surface separation and lubrication regimes**

• No further comments

**Reynolds equation and the EHL lubrication problem**

• No further comments

**Approximation in EHL modeling**

• The authors state in paragraph 140 that "the surface geometries close to the contact region roughly approximate a plane surface", please revise. in the reviewer's opinion this chapter should address the use of a reduced *R*, in which the curvature of both contact bodies is consider in order to allow the use of the aforementioned simplification

**Line and point contacts**

- It seems that not all variables had been introduced. For example, Equation 15 is given without introducing the ellipse axis first
- Please add further information and/or references to the statement in paragraph 180 "other approaches to theses types of equivalence have also been taken in the literature"

**General characteristics of EHL contacts**

- It is commonly known that the film thickness decreases in the area after the PETRUSEVICH-peak. The authors state, furthermore, that this occurs *"in both incompressible and compressible cases"*. Could the authors please describe under which boundary conditions incompressible and compressible cases occur?
- In the case of compressible cases, the authors described that the film thickness is *"slightly"* reduced and the pressure pike *"dramatically"*. Can the authors please quantify the expected decrease? Which case would be the reference case, incompressible behavior?
- Starting at paragraph 230 it becomes unclear whether side-leakage is being consider or not (according to paragraph 215 it is being ignored). Please revise
- Please consider adding further information to Figure 3, which would support the descriptions found in paragraphs 234 and 235

**Dimensionless groupings and film thickness equations**

- Please add references to the presented equations
- Please revise paragraph 275. It seems a bit misplaced
- Paragraph 280: Please try to give a clearer recommendation for the intended readership
- Please give further information regarding the validation strategy and validity range (e.g. oil types) of the reduced analysis given in equations 23 through 25. It is unclear how far this simplified approach can be used in real life applications.

**Accuracy of film thickness equations**

• Paragraph 350: The authors states that according to WEEHLER "the current analytical equations must be considered as providing qualitative, rather than truly quantitative estimates of the film thickness ". Could the authors please comment this statement? There are multiple publications (including in-situ film thickness measurements) that show that the analytical approach can give good results for a wide range of contact conditions, in particular if the oil properties are well known

**Surface roughness interactions**

- On paragraph 425 the authors state that when *"roughness increases, hmin increases.."*. Could the authors comment on this? As described in the first chapters, the commonly used calculation methods for the film thickness do not consider the surface roughness. How does an increase in the surface roughness improves the film build-up?
- On paragraph 435 state that equations 32 and 33 are valid for lambda > 0.5. Please comment on how lambda (especially  $h_{min}$ ) was calculated

**Starvation**

- Paragraph 360: There exist methods to determine the meniscus distance, see:
  - Nogi 2015 (https://doi.org/10.1115/1.4030203)
  - Fischer 2021a (10.1016/j.triboint.2021.106858)
  - o Fischer 2021b (http://dx.doi.org/10.1088/1757-899X/1097/1/012007)

- Chen 2022 (https://doi.org/10.1063/5.0068707)
- Paragraph 370 "film reductions due to starvation depends on bearing operating parameters, especially speed": In the reviewer's opinion further effects (e.g. viscosity and available Oil volume) also play an important role (see for example Fischer 2021b http://dx.doi.org/10.1088/1757-899X/1097/1/012007)

**Grease Lubrication**

- Paragraph 475: I couldn't find the relationship h<cD in Kanazawa 2017. Please specify if this was published or if this is the author's interpretation
- Paragraph 485 "Contact replenishment occurs in grease lubricated bearings, but as a strictly local phenomenon": In the reviewer's opinion this section is quite one-sided (only citing CEN and LUGT). Multiple authors have made significant contributions in this field in the last 50 years. Please consider citing: CANN, ASTRÖM, GONCALVES, FISCHER, POLL, KUHN, HUANG.

**Discussion**

• No further comments. Please consider the points above

**Conclusion**

• In the reviewer's opinion the conclusion is not a conclusion at all

---

## Author Comment (AC1)

*Wind turbine main-bearing lubrication - Part 1: An introductory review of elastohydrodynamic lubrication theory*

**Response to reviewer 1**

Dear reviewer,

Thank you for taking the time to review our manuscript, and for your helpful comments which we will use to improve the quality of this paper. In addition, I must apologise that this response is only being provided now. I (EH) had my first baby arrive in October, and so have been delayed in providing a full response. We include your comments below in **blue**, followed by our responses in **black**.

The manuscript presents a review of Elastohydrodynamic (EHD) theory, starting from the very basic concepts to the application, assumptions and limitations of the theory. It is important to mention that this reviewer understands that the authors try to address the message to a community which might not be too familiar EHD. Thus the reviewer understands that some basic concepts need to stay in the present manuscript, otherwise a more synthetic manuscript could have been written.

Thank you for your understanding regarding the intended audience for this paper, and the implications of this with regards to the contents.

The reviewer is favourable to the publication of the manuscript but requires some minor revisions or at least the answer of some questions.

Revisions:
1. Equation (12): There are increasing voices (Vergne and Bair 2014) arguing that these models are too simplistic to capture the real behaviour of viscosity as a function of pressure and temperature. This is not mentioned in this section although the topic is somehow mentioned in the discussion section.
Thank you for pointing out this discrepancy. When revising the paper we will ensure to provide a discussion of the arguments surrounding viscosity models alongside where Equation 12 is introduced.

2. Equations (26-27): The work of (Habchi and Vergne: https://doi.org/10.1007/s11249-021-01512-z) shows significant deviations among the different authors mentioned here and experimental results for minimum film thickness. Perhaps the authors should comment this.
Thank you for bringing this recent paper to our attention. We agree that it should be included in our discussions. However, please note that it deals specifically with circular contacts (k=1). The poor performance of equations shown in the Habchi and Vergne paper are therefore specific to the circular case. In our paper we focus on the Masjedi and Khonsari equations, but due to their having been shown to perform well for cases of long elliptical contacts (k>1). We also point out in the manuscript that they perform less well than other equations in the circular case. Some of this general context is therefore already included in the manuscript, but the paper you highlight provides new and recent insights and so is relevant to include also.

3. In section: Grease Lubrication, two important references seem to be missing for the understanding of thickener contribution on the EHD film thickness: Nogi, (https://doi.org/10.1080/10402004.2020.1778147) and Morales-Espejel (Tribology International 74 (2014) 7–19), especially for slow rotating bearings.
Thank you for pointing out that these references are missing. We will make sure to introduce and discuss them properly in the revised manuscript.

4. In the Dynamic effects section, the reviewer remarks that starts-stops are not mentioned by the authors, in the targeted application they are multiple and very important, strictly speaking is not a dynamic effect but it is dynamic in nature. Some works have been written on EHL stopping or accelerating contacts with simple formulae.
This is an excellent point, we agree that start-stop events are very relevant to wind turbine bearing operation. We will identify appropriate papers which consider effects related to such events and expand the section on dynamic effects to include them.

Best regards,

Edward Hart
(on behalf of co-authors)

---

## Author Comment (AC2)

Wind turbine main-bearing lubrication - Part 1: An introductory review of elastohydrodynamic lubrication theory

**Response to reviewer 2**

Dear reviewer,

First, we would like to thank you for the considerable effort you have put into assessing our manuscript and suggesting improvements. We will make sure the updated manuscript includes the suggestions you have made.

A detailed response is now provided. We include your comments below in **blue**, followed by our responses in **black**.

The submitted paper gives an overview of the current state of elastohydrodynamic lubrication theory, focusing on simplified descriptions for reaching a wider readership. The authors are encouraged to add further figures in order to support the overarching goal of reaching a wider (non-expert) readership. The complex processes taking place in an EHL contact (e.g. starvation) can be illustrated using existing images. In the current version the reader is challenged to use his power of imagination.
We agree that the paper would benefit from more figures illustrating the various described effects. We will include these in the revised manuscript.

While the depth of the explanations is nicely balanced, the review leaves most of the chapters open. Please consider adding clearer recommendations for the aimed readership.
Since this is part 1 of a two part paper, much of the discussion specific to the main-bearing in wind turbines is handled in part 2. This is where some of this discussion takes place. However, when revising the paper, we will try and include some clear recommendations to the intended audience here as well.

Furthermore, multiple equations are given without proper citation. Therefore, the authors should assess if all references had been included.
We will check to ensure relevant references for all equations are included.

Furthermore, the title is misleading. The overview does not clearly explain the applicability or the relevance of the individual approaches for the case "main-bearing". In the reviewer's opinion the title should be changed or the relevance of each approach (including the validity range) for main-bearing applications should be addressed.
Again, this relates to the fact that this is a two-part paper. This arrangement was agreed with the journal editors prior to submission, but, we will consider whether recommendations can be added to this first part to help bridge the gap between the two.

Please consider also following points:
**Introduction**
• Could the authors please give an overview of other EHL-Reviews (e.g. doi:10.3390/lubricants8050051) and shortly comment the differences between the reviews and the need for a new one?
Excellent suggestion, yes we will do this.

**Surface separation and lubrication regimes**

• No further comments

**Reynolds equation and the EHL lubrication problem**
• No further comments

**Approximation in EHL modeling**
• The authors state in paragraph 140 that *"the surface geometries close to the contact region roughly approximate a plane surface"*, please revise. in the reviewer's opinion this chapter should address the use of a reduced $R$, in which the curvature of both contact bodies is consider in order to allow the use of the aforementioned simplification
We will revise as suggested.

**Line and point contacts**
• It seems that not all variables had been introduced. For example, Equation 15 is given without introducing the ellipse axis first
We will check to make sure all variables are introduced at the appropriate time.

• Please add further information and/or references to the statement in paragraph 180 *"other approaches to these types of equivalence have also been taken in the literature'"*
We will add relevant references here.

**General characteristics of EHL contacts**
• It is commonly known that the film thickness decreases in the area after the PETRUSEVICH-peak. The authors state, furthermore, that this occurs *"in both incompressible and compressible cases".* Could the authors please describe under which boundary conditions incompressible and compressible cases occur?
Compressibility is a characteristic of the lubricant, therefore I am not entirely sure to which boundary conditions you are referring. However, we will consider this point and address it if we are able.

• In the case of compressible cases, the authors described that the film thickness is *"slightly"* reduced and the pressure pike *"dramatically".* Can the authors please quantify the expected decrease? Which case would be the reference case, incompressible behavior?
We will seek to include quantitative numbers to give some indication of relative magnitudes here, likely percentage reductions. Yes, we were using the incompressible case as the reference case.

• Starting at paragraph 230 it becomes unclear whether side-leakage is being consider or not (according to paragraph 215 it is being ignored). Please revise
We will revise as suggested.

• Please consider adding further information to Figure 3, which would support the descriptions found in paragraphs 234 and 235
Excellent suggestion, we will do this!

**Dimensionless groupings and film thickness equations**
• Please add references to the presented equations
Will do!

• Please revise paragraph 275. It seems a bit misplaced

I am not sure if you are referring to the text-box, or the discussion of the dimensionless parameter in the case of a point contact. We will assess this general section for clarity.

• Paragraph 280: Please try to give a clearer recommendation for the intended Readership

I believe this refers to the discussion of optimal dimensionless groupings. I am not sure what recommendation might be given here. Dimensionless groupings are, in general, determined by which film thickness equations is being used. We will consider what might be suggested here though (maybe just that!).

• Please give further information regarding the validation strategy and validity range (e.g. oil types) of the reduced analysis given in equations 23 through 25. It is unclear how far this simplified approach can be used in real life applications.

Excellent suggestion, will do!

**Accuracy of film thickness equations**

• Paragraph 350: The authors states that according to WEEHLER *"the current analytical equations must be considered as providing qualitative, rather than truly quantitative estimates of the film thickness "*. Could the authors please comment this statement? There are multiple publications (including in-situ film thickness measurements) that show that the analytical approach can give good results for a wide range of contact conditions, in particular if the oil properties are well known

This is a good point. I guess the main point to make is that while good results can be obtained in some cases (especially when oil properties are known), in any given applications the quality of results will likely not be known, and so results should be treated as qualitative. This is especially true if oil properties are not well known. We will revise to improve this discussion.

**Surface roughness interactions**

• On paragraph 425 the authors state that when *"roughness increases, hmin increases.."*. Could the authors comment on this? As described in the first chapters, the commonly used calculation methods for the film thickness do not consider the surface roughness. How does an increase in the surface roughness improves the film build-up?

We will revise to try and better describe the interactions here.

• On paragraph 435 state that equations 32 and 33 are valid for lambda > 0.5. Please comment on how lambda (especially $h$min) was calculated

Will do!

**Starvation**

• Paragraph 360: There exist methods to determine the meniscus distance, see:
o Nogi 2015 ( https://doi.org/10.1115/1.4030203)
o Fischer 2021a (10.1016/j.triboint.2021.106858)
o Fischer 2021b (http://dx.doi.org/10.1088/1757-899X/1097/1/012007)
o Chen 2022 (https://doi.org/10.1063/5.0068707)
• Paragraph 370 *"film reductions due to starvation depends on bearing operating parameters, especially speed"*: In the reviewer´s opinion further effects (e.g. viscosity and available Oil volume) also play an important role (see for example Fischer 2021b - http://dx.doi.org/10.1088/1757-899X/1097/1/012007)

Thank you for pointing out this discrepancy. We will improve this section using the literature you suggest here.

**Grease Lubrication**

• Paragraph 475: I couldn't find the relationship h<cD in Kanazawa 2017. Please specify if this was published or if this is the author´s interpretation

We will update the paper to make this point clear.

• Paragraph 485 *"Contact replenishment occurs in grease lubricated bearings, but as a strictly local phenomenon":* In the reviewer's opinion this section is quite one-sided (only citing CEN and LUGT). Multiple authors have made significant contributions in this field in the last 50 years. Please consider citing: CANN, ASTRÖM, GONCALVES, FISCHER, POLL, KUHN, HUANG.

This is a fair point. We will try and provide a more rounded overview of grease results and key contributors, including the authors you suggest.

**Discussion**
• No further comments. Please consider the points above

**Conclusion**
• In the reviewer´s opinion the conclusion is not a conclusion at all

We will revise the conclusion.

Best regards,

Edward Hart
(on behalf of co-authors)

---

## Author Response (AR1)

Wind turbine main-bearing lubrication - Part 1: An introductory review of elastohydrodynamic lubrication theory

**Response to Editor**

Dear Amir,

Thank you for handling the review for our submission to WES. This file includes detailed responses to both reviewers, which have been updated now the requested revisions have been made. The reviewers made excellent suggestions and we have extensively revised the manuscript in line with their comments. This includes new figures to better illustrate the complex mechanisms taking place.

Please note, due to the extent of the changes, the 'track changes' latex file doesn't really work here because it's big chunks of new text rather than scattered shorter edits (the result doesn't aid easy review). Instead we have therefore included a version of the paper wherein the most major edits have been highlighted. I hope this is acceptable, but if not then please let me know.

Best,

Edward Hart
(on behalf of co-authors)

*Wind turbine main-bearing lubrication - Part 1: An introductory review of elastohydrodynamic lubrication theory*

**Updated response to reviewer 1 (post revisions)**

Dear reviewer,

Thank you for taking the time to review our manuscript, and for your helpful comments which we have used to improve the quality of this paper. We include your comments below in **blue**, followed by our updated responses in **black**, having now made the promised edits.

The manuscript presents a review of Elastohydrodynamic (EHD) theory, starting from the very basic concepts to the application, assumptions and limitations of the theory. It is important to mention that this reviewer understands that the authors try to address the message to a community which might not be too familiar EHD. Thus the reviewer understands that some basic concepts need to stay in the present manuscript, otherwise a more synthetic manuscript could have been written.

Thank you for your understanding regarding the intended audience for this paper, and the implications of this with regards to the contents.

The reviewer is favourable to the publication of the manuscript but requires some minor revisions or at least the answer of some questions.

Revisions:
1. Equation (12): There are increasing voices (Vergne and Bair 2014) arguing that these models are too simplistic to capture the real behaviour of viscosity as a function of pressure and temperature. This is not mentioned in this section although the topic is somehow mentioned in the discussion section.

We have now included proper discussion of this in Section 3. The following paragraph has been added "Indeed, there is a growing call for more realistic modelling of lubricant rheological behaviour in general (Bair et al., 2016; Bair, 2019). It is argued that these aspects of EHL must be properly accounted for before it can be considered 120 a truly quantitative discipline (Vergne and Bair, 2014; Bair et al., 2016; Bair, 2019). Despite these shortcomings, there still remain many examples where numerical models employing the above empirical equations are able to accurately recreate results obtained experimentally (e.g. see Tsuha and Cavalca (2020); Venner and Wijnant (2005); Zhang et al. (2020))." (starting on Line 118 of the revised manuscript)

2. Equations (26-27): The work of (Habchi and Vergne: https://doi.org/10.1007/s11249-021-01512-z) shows significant deviations among the different authors mentioned here and experimental results for minimum film thickness. Perhaps the authors should comment this.

As stated in our initial response, the paper you mention is specifically focussed on circular point contacts (k=1). The manuscript already indicates that methods struggle there – stating that errors observed in a Wheeler paper reached around 90%. But we agree it's worth also mentioning this paper and so we have added a footnote to the sentence on errors in circular contacts which reads "[10]Recent work (Habchi and Vergne, 2021) has confirmed that the presented film thickness equations struggle at predicting $h_m$ in point contacts (k = 1). An improved analytical approach is also presented therein." (see Line 371 of the revised manuscript).

3. In section: Grease Lubrication, two important references seem to be missing for the understanding of thickener contribution on the EHD film thickness: Nogi, (https://doi.org/10.1080/10402004.2020.1778147) and Morales-Espejel (Tribology International 74 (2014) 7–19), especially for slow rotating bearings.

The grease section has now been overhauled as a result of the improvements to the starvation section. Thank you for pointing out our missing these two important additional papers. They are both now discussed in the updated manuscript in the grease section.

4. In the Dynamic effects section, the reviewer remarks that starts-stops are not mentioned by the authors, in the targeted application they are multiple and very important, strictly speaking is not a dynamic effect but it is dynamic in nature. Some works have been written on EHL stopping or accelerating contacts with simple formulae.

This is an excellent point! We agree that start-stop events are very relevant to wind turbine bearing operation. We have therefore identified and discussed key effects and appropriate papers which consider such effects. Please see lines 582-599 in the updated manuscript.

Best regards,

Edward Hart
(on behalf of co-authors)

Wind turbine main-bearing lubrication - Part 1: An introductory review of elastohydrodynamic lubrication theory

**Updated response to reviewer 2 (post revisions)**

Dear reviewer,

Thank you again for the considerable effort you have put into assessing our manuscript and suggesting improvements.

A detailed response is now provided. We include your comments below in **blue**, followed by our responses in **black**, having now made the promised edits.

The submitted paper gives an overview of the current state of elastohydrodynamic lubrication theory, focusing on simplified descriptions for reaching a wider readership. The authors are encouraged to add further figures in order to support the overarching goal of reaching a wider (non-expert) readership. The complex processes taking place in an EHL contact (e.g. starvation) can be illustrated using existing images. In the current version the reader is challenged to use his power of imagination.
We agree that the paper would benefit from more figures illustrating the various described effects. We have therefore generated new figures which better describe (visually) the following: Hertzian contact stress distributions (Fig 2), starvation (Figs 5 and 6), non-steady effects in EHL (Fig 7). Thank you for this excellent suggestion.

While the depth of the explanations is nicely balanced, the review leaves most of the chapters open. Please consider adding clearer recommendations for the aimed readership.
Since this is part 1 of a two-part paper, much of the discussion specific to the main-bearing in wind turbines is handled in part 2. This is where much of this discussion takes place. When revising we have however tried to make recommendations clearly where relevant, for example in the revised conclusion of the updated manuscript.

Furthermore, multiple equations are given without proper citation. Therefore, the authors should assess if all references had been included.
We have been back through the manuscript and have added citations where they were lacking.

Furthermore, the title is misleading. The overview does not clearly explain the applicability or the relevance of the individual approaches for the case "main-bearing". In the reviewer's opinion the title should be changed or the relevance of each approach (including the validity range) for main-bearing applications should be addressed.
Again, this relates to the fact that this is a two-part paper. This arrangement was agreed with the journal editors prior to submission and we still believe that to be the most useful form for this work to take.

Please consider also following points:
**Introduction**
• Could the authors please give an overview of other EHL-Reviews (e.g. doi:10.3390/lubricants8050051) and shortly comment the differences between the reviews and the need for a new one?
This has been added to the introduction as requested.

**Surface separation and lubrication regimes**
• No further comments

**Reynolds equation and the EHL lubrication problem**
• No further comments

**Approximation in EHL modeling**
• The authors state in paragraph 140 that *"the surface geometries close to the contact region roughly approximate a plane surface"*, *p*lease revise. in the reviewer's opinion this chapter should address the use of a reduced *R*, in which the curvature of both contact bodies is consider in order to allow the use of the aforementioned simplification

The use of a reduced R allows the problem to be reformulated as that of an ellipsoid approaching a flat plane, but in addition the stated approximation is only valid if contact dimensions are small relative to the reduced R values. Therefore, reducing the problem using reduced radii isn't the only step here. However, we agree that R values could be better introduced and so we have added the sentence "The problem of two curved bodies in contact can be reduced to that of a single ellipsoid or cylinder contacting a flat plane (Harris and Kotzalas, 2006b). The geometry of the equivalent ellipsoid or cylinder is captured by the reduced radii of curvature Rx and Ry in x and y directions (see Appendix A)." Later in that same paragraph we have retained the sentence "The half-space approximation is valid only if: surface geometries close to the contact region roughly approximate a plane surface,…" but please note that this point is further clarified is the sentence "These requirements are satisfied if the significant dimensions of the contact region are small with respect to the dimensions of the contacting bodies and the relative radii of curvature of the surfaces (Johnson, 1987)."

**Line and point contacts**
• It seems that not all variables had been introduced. For example, Equation 15 is given without introducing the ellipse axis first

This has been rectified and the manuscript checked to try and ensure all variables are introduced at the appropriate time.

• Please add further information and/or references to the statement in paragraph 180 *"other approaches to these types of equivalence have also been taken in the literature'"*

This paragraph has been updated and now reads "Other approaches to these types of equivalence have been taken in the literature; for example, seeking an equivalent line contact in which the maximum or mean Hertzian pressure coincides with that of the point contact for cases where patch widths (b) in the line and point contact don't coincide (Hamrock and Dowson, 1981)."

**General characteristics of EHL contacts**
• It is commonly known that the film thickness decreases in the area after the PETRUSEVICH-peak. The authors state, furthermore, that this occurs *"in both incompressible and compressible cases".* Could the authors please describe under which boundary conditions incompressible and compressible cases occur?

Compressibility is a characteristic of the lubricant; therefore, we are not sure to which boundary conditions you are referring. We have looked through relevant literature and have been unable to work out what you were referring to here. Please feel free to confirm and we'll be happy to try and accommodate your suggestion.

• In the case of compressible cases, the authors described that the film thickness is *"slightly"* reduced and the pressure pike *"dramatically".* Can the authors please quantify the expected decrease? Which case would be the reference case,

We have edited this paragraph as follows "In practise, it has been found that a marked reduction in film thickness occurs close to the outlet, as shown in Figure 2, in both incompressible and compressible cases. In the latter case the pressure spike magnitude is dramatically reduced relative to incompressible results (one example in Hamrock et al. (2004) sees a reduction of 3.7 times). The central film thickness, hc, is smaller (on the order of tens of %) for compressible flow under otherwise identical conditions, while the minimum film thickness, hm, only changes by a few percent (Venner and Bos, 1994; Hamrock et al., 2004)." Please see line 239 of the revised manuscript.

• Starting at paragraph 230 it becomes unclear whether side-leakage is being consider or not (according to paragraph 215 it is being ignored). Please revise

Side leakage is only ignored in the simplified illustrative case used as an example to motivate why film height reductions occur. Everything which follows is observed with side-leakage included. We agree this need to be clearer, as such we have made the following edits: "We briefly consider the simplified case in which flow in the y-direction (side-leakage) is ignored;… In practise, and both with or without side leakage, it has been found that…". Then, when discussing the horseshoe shape in the point contact the presence of side leakage is explicitly mentioned. We believe this now clears up any possible confusion for the reader.

• Please consider adding further information to Figure 3, which would support the descriptions found in paragraphs 234 and 235

Example local lubricant flow velocities have been added to this figure (now Figure 4) to better represent what is described in the text.

**Dimensionless groupings and film thickness equations**

• Please add references to the presented equations

This has now been done.

• Please revise paragraph 275. It seems a bit misplaced

My interpretation is that maybe you are suggesting that the description of ellipticity ratio is misplaced here, since it feels more related to the Hertzian contact section? It is mentioned here also because it is one of the dimensionless parameters which characterises the contact along with W, U and G. In addition we also mention the dimensionless film thickness in the same paragraph also for this reason. Generally people focus on W, U and G but those are not enough alone and kappa and H are also needed.

• Paragraph 280: Please try to give a clearer recommendation for the intended Readership

I believe this refers to the discussion of optimal dimensionless groupings. I am not sure what recommendation might be given here. Dimensionless groupings are, in general, determined by which film thickness equations is being used.

• Please give further information regarding the validation strategy and validity range (e.g. oil types) of the reduced analysis given in equations 23 through 25. It is unclear how far this simplified approach can be used in real life applications.

On reconsidering this comment we believe there might be a misunderstanding here regarding these optimal non-dimensional parameters. As discussed in the manuscript, dimensional analysis is applied to a system of mathematical equations in order to identify optimal dimensional reduction and final parameters within that system. More specifically, the techniques for identifying Equations 23-25 are exact and so the presented parameters are exactly the optimal set for the EHL equations as defined in Section 3 of the paper. Whether or not those equations represent what is happening in the real world is essentially a separate matter, and one which we tackle principally when considering the

accuracy of identified analytical equations for the film thickness. But you raise the good point that it wasn't entirely clear that these parameters are optimal for the EHL problem *as defined in Section 3* and so confusion could well arise here. To try and make this point clearer we have now edited this paragraph to read *"For the EHL problem (as defined by equations in Section 3) in line and points contacts,…"*. With regard to different lubricant types we also clarify further down "Additional parameters are required to fully capture more complex viscosity and density characteristics, for full details see Hsiao (2001).

**Accuracy of film thickness equations**
• Paragraph 350: The authors states that according to WEEHLER *"the current analytical equations must be considered as providing qualitative, rather than truly quantitative estimates of the film thickness "*. Could the authors please comment this statement? There are multiple publications (including in-situ film thickness measurements) that show that the analytical approach can give good results for a wide range of contact conditions, in particular if the oil properties are well known

While good results can be achieved, we believe the conclusion of the Wheeler paper is that one cannot generally know when that is the case, and if it isn't then the results may be quite off. For this reason, they suggest current equations are qualitative rather than truly quantitative. However, you make an excellent point and so we have revised this discussion to also point out that good results can still often be achieved using existing film equations so long as lubricant properties are well known. This helps provide a more rounded view of this aspect of EHL and so is a useful addition to the discussion.

**Surface roughness interactions**
• On paragraph 425 the authors state that when *"roughness increases, hmin increases.."*. Could the authors comment on this? As described in the first chapters, the commonly used calculation methods for the film thickness do not consider the surface roughness. How does an increase in the surface roughness improves the film build-up?

Surface roughness is essentially perturbations in film height (h) values locally. These then influence terms in the Reynolds equation. In addition, the squeeze film term (dh/dt) means that the flow has 'memory' of what's gone before, meaning effects are cumulative in some sense. Furthermore, note that the Reynolds equation has h^3 terms present. This means that the effect of a local increase in h is different to that of an equal decrease. In addition, with roughness present, some of the load is carried by contacting surface asperities and not the lubricant flow. This all means that, even under homogenous roughness, the mean film level will be different to that for a smooth surface. Properly solving under these conditions actually requires a reformulation of the Reynolds equation (introducing so called 'flow factors' which introduce stochastic elements that capture the mean effects we're describing). It is the solutions of these modified Reynolds equations that show us the increase in hm that we describe. Hopefully my hand-wavy description above has given you some idea of what underlies that result. Yes, most film thickness equations assume smooth surfaces. Note that (as discussed in this section), it is appropriate to assume smooth surfaces for small enough roughness levels – the dimensionless values associated with which we have stated. But, for rougher surfaces the adjustment factor indicated should be used, yes.

• On paragraph 435 state that equations 32 and 33 are valid for lambda > 0.5. Please comment on how lambda (especially *h*min) was calculated

The study used to determine the starvation correction factors implemented an average flow model of the Reynolds equations (as described in the previous comment) in order to account for the mean effect of roughness interactions. This model (developed by Patir and Cheng in 1978) contains terms (flow factors) which are only valid for Lambda values greater than 0.5. The stated restriction is therefore based on the development of the implemented model itself and not testing occurring later

on. In this setting Lambda was therefore known exactly, since both sigma and hm were available to the model developers.

**Starvation**
• Paragraph 360: There exist methods to determine the meniscus distance, see:
o Nogi 2015 ( https://doi.org/10.1115/1.4030203)
o Fischer 2021a (10.1016/j.triboint.2021.106858)
o Fischer 2021b (http://dx.doi.org/10.1088/1757-899X/1097/1/012007)
o Chen 2022 (https://doi.org/10.1063/5.0068707)
• Paragraph 370 *"film reductions due to starvation depends on bearing operating parameters, especially speed"*: In the reviewer´s opinion further effects (e.g. viscosity and available Oil volume) also play an important role (see for example Fischer 2021b
- http://dx.doi.org/10.1088/1757-899X/1097/1/012007)

We have extensively reworked the starvation section to include this more recent work and an overall much improved characterisation of this operating state. This includes new figures as well. Thank you you for pointing out how lacking our first version was compared to the current state of knowledge.

**Grease Lubrication**
• Paragraph 475: I couldn't find the relationship h<cD in Kanazawa 2017. Please specify if this was published or if this is the author´s interpretation

This is our interpretation/paraphrasing of their findings and discussion points. They make the most directly similar statement in their conclusions. We have added the following footnote to the manuscript to clarify "[13]This is our interpretation of Kanazawa et al. (2017) results which summarises and paraphrases their findings and proposed mechanisms of grease film formation." Please see line 664 of the revised manuscript.

• Paragraph 485 *"Contact replenishment occurs in grease lubricated bearings, but as a strictly local phenomenon":* In the reviewer's opinion this section is quite one-sided (only citing CEN and LUGT). Multiple authors have made significant contributions in this field in the last 50 years. Please consider citing: CANN, ASTRÖM, GONCALVES, FISCHER, POLL, KUHN, HUANG.

We have reworked the grease section of the paper to try and better represent the various contributions. In addition, the changes to the starvation section also required follow-through changes here as well. We believe we now provide a more balanced view of this topic.

**Discussion**
• No further comments. Please consider the points above

**Conclusion**
• In the reviewer´s opinion the conclusion is not a conclusion at all

The discussion and conclusion have now been combined, as we agree that the previous conclusion served no great purpose. The text itself has also been updated.

Best regards,

Edward Hart
(on behalf of co-authors)